

# A year in the life of a central California kelp forest: physical and biological insights into biogeochemical variability

David A. Koweek[1,*], Kerry J. Nickols[2,3], Paul R. Leary[2], Steve Y. Litvin[2], Tom W. Bell[4], Timothy Luthin[1], Sarah Lummis[2,#], David A. Mucciarone[1], and Robert B. Dunbar[1]

[1]Department of Earth System Science, Stanford University, Stanford, CA 94305, USA
[2]Department of Biology, Hopkins Marine Station, Stanford University, Pacific Grove, CA 93950, USA
[3]School of Natural Sciences, California State University Monterey Bay, Seaside, CA 93955, USA
[4]Earth Research Institute, University of California, Santa Barbara, CA 93106, USA
[*]present address: Department of Global Ecology, Carnegie Institution for Science, Stanford, CA 94305, USA
[#]present address: Department of Ecology and Evolutionary Biology, University of California Santa Cruz, Santa Cruz, CA 95060, USA

*Correspondence to*: David A. Koweek (dkoweek@carnegiescience.edu)

**Abstract.**

Kelp forests are among the world's most productive marine ecosystems, yet little is known about their biogeochemistry. This study presents a fourteen-month time series (July 2013-August 2014) of surface and benthic dissolved inorganic carbon and total alkalinity measurements, along with accompanying hydrographic measurements, from six locations within a central California kelp forest. We present ranges and patterns of variability in carbonate chemistry, including pH (7.70-8.33), $pCO_2$ (172-952 $\mu$atm), and the aragonite saturation state, $\Omega_{Ar}$ (0.94-3.91). Surface-to-bottom gradients in $CO_2$ system chemistry were as large as the spatial gradients throughout the bottom of the kelp forest. Dissolved inorganic carbon variability was the main driver of the observed $CO_2$ system variability. The majority of spatial variability in the kelp forest can be explained by advection of cold, dense high $CO_2$ waters into the bottom of the kelp forest, with deeper sites experiencing high $CO_2$ conditions more frequently. Despite the strong imprint of advection on the biogeochemical variability of the kelp forest, surface waters were undersaturated with $CO_2$ in the spring through fall, indicative of the strong role of photosynthesis on biogeochemical variability. We emphasize the importance of spatially distributed measurements for developing a process-based understanding of kelp forest ecosystem function in a changing climate.

## 1. Introduction

Kelp forests are found along rocky coastlines in temperate-to-sub-polar coastal regions throughout the world's oceans. Kelp are important foundation species that support diverse biological communities, including invertebrates, fishes, and marine mammals through their creation of complex, three-dimensional biological habitat and the provisioning of carbon and nutrients which magnify secondary production in the coastal zone (Steneck et al., 2002). Kelp forest ecology has been a



focus of study since the early twentieth century, largely due to its importance as an ecologically rich habitat (as reviewed in Graham et al. 2007).

Kelp (including *Ecklonia spp., Laminaria spp., and Macrocystis spp.*) forests are among the most productive marine ecosystems (Mann, 1982), with linear extension rates of kelp fronds that can range from 2-14 cm d$^{-1}$ (Graham et al., 2007).

Productivity rates vary depending upon the geographic region and species of consideration, but generally range from 600-1500 g C m$^{-2}$ yr$^{-1}$ (Mann, 1982), although they have been documented to reach up to 3400 g C m$^{-2}$ yr$^{-1}$ (Jackson, 1977). The high productivity in kelp forests impacts the chemical properties of the surrounding water through the uptake and release of dissolved inorganic carbon (DIC) and dissolved nutrients (nitrate and phosphate). Calcification and dissolution modify the water chemistry through the uptake and release of carbonate and bicarbonate ions, which modify the total alkalinity (TA)

and DIC. Air/sea gas exchange further modifies the DIC through exchange of $CO_2$. In addition, in the upwelling zones that support kelp forests, carbon system chemistry undergoes large fluctuations due to highly dynamic regional-scale advection of high $CO_2$ deep waters (Booth et al., 2012). The combination of biological and physical processes can lead to large biogeochemical variability in kelp forests. For example, pH$_{NBS}$ values of up to 9.1 have been recorded in one sub-Antarctic kelp forest while matching samples taken directly outside the kelp bed had pH$_{NBS}$ of 8.3 (Delille et al., 2000). In southern

California, Frieder et al. (2012) measured pH$_T$ values as low as 7.7 (corresponding to pCO$_2$ of 1000 μatm).

Despite kelp forest's recognized importance as an ecosystem architect and biogeochemical agent, little attention has been paid to biogeochemical variability in kelp forests (see Delille et al., 2000, 2009; Frieder et al., 2012; Hofmann et al., 2011; Kapsenberg and Hofmann, 2016; Takeshita et al., 2015). Of the few studies that have addressed kelp forest biogeochemistry, most have focused on pH measurements in southern California (Frieder et al., 2012; Hofmann et al., 2011;

Kapsenberg and Hofmann, 2016; Takeshita et al., 2015), which while certainly useful, does not provide complete information about the $CO_2$ system chemistry in kelp forests. The lack of geographically distributed biogeochemical observations in kelp forests may be due to the difficulties of long-term field observations and/or the lack of automated instrumentation and procedures for making continuous, high-quality biogeochemical measurements until recently (Bresnahan et al., 2014). Our limited understanding of kelp forest $CO_2$ system chemistry inhibits our ability to establish

biogeochemical baselines as well as to forecast how kelp forest communities may interact with changes in water chemistry from local stressors, such as runoff and sedimentation, as well as from global stressors, such as climate change and ocean acidification (OA).

The genus *Macrocystis* (order Laminariales), or giant kelp, is the most widely distributed kelp genus in the world. Found in both the northern and southern hemispheres, it dominates kelp assemblages in southern and central California

(Graham et al., 2007). We present results from a 14-month biogeochemical study in a central California *Macrocystis pyrifera*-dominated kelp forest. Our goals were to quantify temporal and spatial carbon system biogeochemical variability in a kelp forest over an annual cycle and investigate the responsible mechanisms.



## 2. Methods

### 2.1 Study Site

Our study site is located along the eastern side of the Monterey Peninsula in the central portion of the California Current Large Marine Ecosystem (Fig. 1) (Checkley and Barth, 2009). This region is characterized by seasonal upwelling

conditions driven by increased northwesterly winds from ~March-September (Checkley and Barth, 2009) (Fig. S1). Although much of the Monterey Peninsula is protected from the stronger along shore winds and associated surface conditions experienced by the nearby exposed coastline and northern Monterey Bay, it still experiences cross-shore transport associated with upwelling (Woodson, 2013). Advection of upwelled waters and the local topography also facilitate the propagation of internal bores into southern Monterey Bay, which act as an additional mechanism for introducing dense, high

$CO_2$ deep ocean water into nearshore habitats and driving variability in $CO_2$ chemistry at short temporal scales (Booth et al., 2012; Walter et al., 2014).

The kelp forest is located in the Lovers Point-Julia Platt State Marine Reserve (LPJPSMR), directly offshore of Hopkins Marine Station along the wave-protected side of the Monterey Peninsula (Fig. 1). LPJPSMR was created in 2007 as part of the network of central California marine protected areas designated under the California Marine Life Protection Act

(California Department of Fish and Wildlife 2016). The kelp forest in LPJPSMR has been protected since 1931, originally as the Hopkins Marine Life Refuge. Due to its protected status, LPJPSMR serves as a natural laboratory to investigate the biogeochemistry of a central California kelp forest featuring low levels of human disturbance.

Within this kelp forest we selected six sites that span gradients in wave exposure, depth, and proximity to the kelp bed (Fig. 1; referenced as Protected Inside, Protected Middle, Protected Offshore, Exposed Middle, Exposed Offshore, and

KFA). Sites inside the kelp bed ranged from 7-11 m and sites just outside the kelp bed ranged from 13-16 m depth. The site inside the kelp bed on the wave-exposed side (Exposed Middle) was slightly deeper than sites on the protected side because we were unable to reliably sample the wave-exposed side of the kelp bed in shallower waters for operational reasons. The six sites ranged from 130-270 m offshore (Table 1). Sites along the transect on the western side of kelp forest experienced greater wave exposure than sites on the protected side due to the northwestern directionality of incoming waves (Fig. S2).

Both the wave-exposed and wave-protected transects were oriented perpendicular to the kelp forest in order to sample within and just outside of the kelp bed. The KFA site (located above the main node of the Kelp Forest Array, an underwater cabled observatory) was located east of the protected transect and served as a kelp-free control site, although it was close enough to the kelp forest to be influenced by the advection of chemically-modified water from the kelp forest.

### 2.2 Water Sampling

We sampled biogeochemical and hydrographic properties at the six study sites at approximately weekly intervals spanning July 2013-August 2014. Samples were collected between 0900 and 1200 hrs in order to minimize any potential



confounding influence of the diel cycle on the weekly-scale biogeochemical variability. We collected water samples for DIC and TA analysis at 1 m below the surface (1 mbs) and 1 m above the bottom (1 mab) using 1.7L Niskin bottles at all 6 sites (Fig 1). Water was sampled from the Niskin bottle into a 30mL bottle for DIC and a 300mL bottle for TA. Samples were immediately preserved by addition of 30 μL (DIC) or 300 μL (TA) saturated mercuric chloride solution ($HgCl_2$) upon

collection from Niskin bottles. Samples were typically analyzed within 1-2 weeks of collection. We conducted simultaneous CTD hydrocasts (SBE 19plus, SeaBird Electronics, Inc.) at each station to characterize the vertical water column structure and provide *in-situ* temperature and salinity necessary for carbon system calculations (see below).

We used CTD hydrocasts to quantify water column stratification by determining the density difference the between the upper two meters and the bottom two meters.

$$\Delta\rho = \overline{\rho}_{bottom} - \overline{\rho}_{surface} \qquad (1)$$

where $\Delta\rho$ is in kg m$^{-3}$, $\overline{\rho}_{bottom}$ is the average density across the bottom two meters of the water column, and $\overline{\rho}_{surface}$ is the average density across the top two meters of the water column. We chose to average across two meters at both the top and bottom sites in order to span the depth of water sample collection (1 mbs and 1 mab).

Sample sets from July and August 2013 were collected over two days, with the samples from the protected transect

typically collected a day before the samples from the exposed transect and the KFA site. We include these first two months of the time series when presenting the data as time series, but in order to avoid confounding variability from sampling on different days, any of the analyses presented here which require comparisons between the sites are limited to the period of September 2013-August 2014 when samples were all collected within approximately 90 minutes of one another.

**2.3 Water chemistry**

Water samples were analyzed at the Stanford University Stable Isotope Biogeochemistry Laboratory (Stanford, California, USA). DIC was measured with a custom-built sample acidification and delivery system coupled to an infrared gas analyzer (Licor 7000) described in Long et al. (2011). The instrument was calibrated prior to each use and monitored throughout run sequences with Certified Reference Materials (CRM) provided by A. Dickson (Scripps Institution of Oceanography). Instrumental precision from 467 CRM analyses over the length of the study was ±1.9 μmol kg$^{-1}$ (1 S.D.).

Immediate duplicate analyses of samples usually yielded instrumental precision of 1-2 μmol kg$^{-1}$.

Samples for TA measurements were pre-filtered through a 0.45 μm filter before being analyzed on a Metrohm 855 Robotic Titrosampler (Metrohm USA, Inc.) using certified 0.1N HCl provided by A. Dickson (Scripps Institution of Oceanography). Total alkalinity calculations from raw titration data follow Dickson et al. (2003). Sample runs were corrected based on the offset between the measured and certified value for CRMs, resulting in an accuracy that was always

better than 1.7% and was better than 0.5% on 79% of analyses. Instrumental precision from 272 CRM analyses over the





duration of this study was ±2.2 µmol kg⁻¹ (1 S.D.). Immediate duplicate analyses of samples usually yielded instrumental precision of 1-2 µmol kg⁻¹.

**2.4 Carbon system calculations**

We calculated pH, pCO$_2$, and Ω$_{Aragonite}$ (Ω$_{Ar}$) in CO2SYS (van Heuven et al., 2011) using DIC, TA, *in-situ*
temperature, and salinity data. We assumed silica concentrations of 3 µmol kg⁻¹ (Brzezinski et al., 2003) and phosphate concentrations of 0.5 µmol kg⁻¹ and 1 µmol kg⁻¹ for the surface and bottom, respectively. We used the carbonate system dissociation constants from Mehrbach et al. (1973) as re-fit by Dickson and Millero (1987), and K$_{SO4}$ for the bisulfate ion from Dickson (1990). pH data is reported here on the total hydrogen ion scale (pH$_T$) at in-situ temperature.

We estimated error for pH, pCO$_2$, and Ω$_{Ar}$ calculations using a Monte Carlo approach. We randomly selected ten
samples in the data set, used the above-listed instrumental precisions for DIC and TA, assumed no error on the temperature and salinity measurements, and performed 1000 iterations on each of the ten samples. We did not consider error on the carbonate system equilibrium constants. The Monte Carlo simulations resulted in simulated distributions of pH, pCO$_2$, and Ω$_{Ar}$ for each of the ten randomly selected samples in the data set. We calculated the standard deviation of each distribution. The maximum standard deviation of the set of ten simulated distributions of pH, pCO$_2$, and Ω$_{Ar}$ are 0.01 units, 20 µatm, and,
0.03 units, respectively. We consider these error estimates to be conservative since they are the maximum, instead of average, values.

We quantified the effects of TA, DIC, temperature, and salinity on the observed vertical differences ($\Delta y_{Top-Bottom}$) in pH, pCO$_2$, and Ω$_{Ar}$, using a first order Taylor series budget following Hauri et al. (2013):

$$\Delta y_{Top-Bottom} = \frac{\partial y}{\partial T}\Delta T_{T-B} + \frac{\partial y}{\partial S}\Delta S_{T-B} + \frac{\partial y}{\partial TA}\Delta TA_{T-B} + \frac{\partial y}{\partial DIC}\Delta DIC_{T-B} \qquad (2)$$

where $y$ is pH, pCO$_2$, or Ω$_{Ar}$ and the terms on the right-hand-side of the equation account for the effects of temperature,
salinity, TA, and DIC, respectively. The partial derivatives ($\frac{\partial y}{\partial T}, \frac{\partial y}{\partial S}, \frac{\partial y}{\partial TA}, \frac{\partial y}{\partial DIC}$) were estimated numerically in
CO2SYS centered at the mean values of temperature, salinity, TA, and DIC across the entire data set (Table S1).

**2.4 Satellite-derived kelp canopy biomass estimates**

We estimated giant kelp canopy biomass from July 2013 to August 2014 using multispectral Landsat 7 Enhanced
Thematic Mapper and Landsat 8 Operational Land Imager imagery (Bell et al., 2015a; Cavanaugh et al., 2011). Briefly, each Landsat image was atmospherically corrected using 50 temporally stable pseudo-invariant targets to standardize radiometric signals across dates. The proportion of kelp canopy in each 30m x 30m pixel was determined using multiple end-member spectral mixing analysis (Roberts et al., 1998), in which each pixel was modeled as a combination of 1 static kelp end-



member and 30 seawater end-members, which were unique to each image. Kelp canopy biomass was estimated using the observed relationship between diver estimated canopy biomass and Landsat pixel kelp fraction in the Santa Barbara Channel (where this technique was originally applied-see Cavanaugh et al. (2011)).

### 2.4 Current velocity time series

Water-column velocity was measured continuously near the exposed and protected sites in the middle of the kelp forest using bottom-mounted Acoustic Doppler Current Profilers (RD Instruments, 1200 kHz). The instruments recorded ensemble averaged velocities every 3 minutes in 0.5 m bins extending from ~1.5 m above the bottom to ~1.5 m below the surface. The instrument on the protected side of the kelp forest was deployed throughout this study, while the instrument on the exposed side of the kelp forest was deployed through December 2013.

## 3. Results

### 3.1 Kelp canopy biomass

Satellite-estimated kelp canopy biomass displayed a strong seasonal cycle, consistent with previous observations of the strong seasonality of kelp canopy biomass in central California (Bell et al., 2015b; Reed et al., 2011). Canopy biomass increased throughout spring, reached a maximum during the summer months, and decreased throughout the fall, reaching

minimum values in the winter. The timing of kelp growth, senescence, and canopy biomass range were generally consistent between the wave-protected and wave-exposed sides of the kelp forest (Fig. 2a). However, canopy biomass on the exposed side began to decline slightly before canopy biomass on the protected side in summer 2014, leading to a decoupling of canopy biomass between the two sides from June-September 2014.

### 3.2 Water column structure

The kelp forest water column was strongly influenced by seasonal variations in surface and bottom water temperature. We use the protected side transect, which spans the largest range in site depths, to examine these processes in detail (Fig. 3). During the spring, cold (~10°C), dense bottom water was advected into the deepest kelp forest site (Protected Offshore), causing water column temperature differences of up to 4°C. This cold bottom water did not reach further inshore along the bottom of the shallower, inwards Protected Middle and Protected Inside sites (Fig. 3). Surface heating during

summer 2014 resulted in uniformly warm surface waters (~16-18°C) across the sites along the protected transect. Reduced stratification at the shallower Protected Inside site allowed warmer surface waters to mix downwards, which resulted in a nearly isothermal water column whereas temperature stratification was maintained in the deeper Protected Offshore site. Salinity showed a seasonal cycle across all three sites of the protected transect with higher salinity during upwelling season, where values ranged from ~33.6-34, as compared to less saline winter months, where salinity ranged from 33-33.6 (Fig. S3).





However, there was little evidence for depth-dependent variation in salinity, suggesting that temperature was the dominant control on water column stratification. These patterns were similar across all sites.

Water column stratification, as quantified by $\Delta\rho$, was greatest and most variable from the start of upwelling season throughout the fall where $\Delta\rho$ exceeded 1 kg m$^{-3}$, with strong site-to-site variability (e.g., June-August 2014; Fig. 2b). Throughout most of the year, $\Delta\rho$ was greatest and most variable at the Protected Offshore, KFA, Exposed Offshore, and Exposed Mid sites. Protected Middle and Protected Inside typically, although not always, had lower $\Delta\rho$ and less variability from week-to-week. All sites were minimally stratified in the winter months, as $\Delta\rho$ generally stayed below 0.4 kg m$^{-3}$ and was generally less than 0.2 kg m$^{-3}$ in December 2013-February 2014. Site-to-site variability was also reduced in the winter months, although there was one anomalously high stratification observation at the Exposed Offshore site in early January 2014, where $\Delta\rho$ nearly reached 1 kg m$^{-3}$.

### 3.3 CO$_2$ system chemistry time series

The 14-month time series of DIC for all six sites near the surface (1 m below sea level) and the bottom (1 m above the sea floor) exhibited strong week-to-week, site-to-site, and seasonal variability (Fig. 4a). Bottom DIC frequently exceeded 2100 μmol kg$^{-1}$ during Fall 2013 and exceeded 2200 μmol kg$^{-1}$ at all sites except Protected Middle during April-May 2014. The deeper sites typically had highest bottom DIC during the spring and summer months, although the site-to-site variability disappeared during the winter. Surface DIC concentrations were generally much more spatially homogeneous than bottom water DIC concentrations. Surface DIC reached minimum values during spring and summer months when DIC was frequently less than 2000 μmol kg$^{-1}$.

The time series of surface and bottom TA shows reduced variability compared to the DIC time series (Fig. 4b). Surface and bottom TA samples generally ranged from 2240 to 2260 μmol kg$^{-1}$, with occasional higher values observed. Unlike DIC, TA does not show any consistent surface-to-bottom gradients throughout the time series. Increased TA during the upwelling season was likely due to the high nutrient content of the source waters being advected into the kelp forest.

Time series of pH, pCO$_2$, and $\Omega_{Ar}$ all exhibited similar properties to the DIC time series with strong week-to-week, site-to-site, and seasonal variability (Fig. 5). Minimum pH values in bottom samples ranged from 7.70-7.79 across the six sites, and pH minima grouped closely to site depth, where the four deeper sites has pH minima which ranged from 7.70-7.72, while the pH minima at Protected Middle was 7.78 and the pH minima at Protected Inside was 7.79 (Fig. 5a). Bottom $\Omega_{Ar}$ values exhibited similar clustering as pH. Surface pCO$_2$ was undersaturated (<400 μatm) through the early fall, spring, and summer, although week-to-week variability resulted in periods of pCO$_2$ saturation and supersaturation during July and August 2014 (Fig. 5b). We observed $\Omega_{Ar}$ undersaturation ($\Omega_{Ar}<1$) five times: at Protected Offshore on 25 April 2014 and 14 May 2014 and at KFA, Exposed Offshore, and Exposed Middle on 14 May 2014. Table 2 provides a summary of all carbon system variable ranges.





### 3.4 Surface-to-bottom variability

The combination of maximum kelp abundance and water column stratification led to greatest carbon system variability within the kelp forest during the upwelling season (~March-September). Ranges in DIC along the top, along the bottom, and vertical gradients at each sampling site, exceeded 100 µmol kg$^{-1}$ at this time (Fig. 6a). Winter mixing from

storms and the lack of kelp canopy reduced these differences, resulting in chemical gradients which approached 0 µmol kg$^{-1}$. TA spatial variability was much smaller and more variable (Fig. 6b). pH, pCO$_2$, and Ω$_{Ar}$ exhibited strong top to bottom differences as well. Vertical differences in pH and Ω$_{Ar}$ reached up to 0.51 and 2.25 units, respectively.

Nearly all of the surface-to-bottom variability in pH, pCO$_2$ and Ω$_{Ar}$ can be attributed to vertical differences in DIC using Eq. (2). The DIC contribution to the observed surface-to-bottom differences in pH, pCO$_2$ and Ω$_{Ar}$ across the six sites

ranged from 88-93%, 82-88%, and 92-97%, respectively (Fig. 7 and S4-6). These results are consistent with an ecosystem dominated by organic carbon metabolism (photosynthesis/respiration) as opposed to inorganic carbon metabolism (calcification/dissolution), an expected result in a kelp forest. We note that the carbon system budgets do not discriminate between local biogeochemical modification within the kelp forest and advection of offshore water.

Cold, high DIC water observed at the bottom of the kelp forest was a major contributor to surface-to-bottom

differences in DIC (Fig. 8). High DIC water reached the deeper sites (Protected Offshore, KFA, Exposed Offshore, Exposed Middle), but did not frequently penetrate into the shallower sites (Protected Middle and Protected Inside). Bottom DIC accounted for 55-67% (pearson correlation coefficients) of observed surface-to-bottom DIC gradients in the four deeper sites (Fig. 8b), but only accounted for 39% of the observed gradient at Protected Middle and 22% of the gradient at Protected Inside. Reduced stratification at the protected shallower sites reduced water column temperature and DIC gradients (Figs. 2b,

3, and 4).

The vertical variability in CO$_2$ system chemistry drove large variations in the ability to buffer against ongoing ocean acidification. The Revelle Factor describes the non-linear buffering capacity of seawater $RF = \partial \ln pCO_2 / \partial \ln DIC$. Higher *RFs* indicate reduced capacity to buffer against increasing CO$_2$ as ocean acidification progresses. *RF* variability within the kelp forest ranged from ~9-18 over the annual cycle (Fig. S7), which nearly spans the range of global surface

ocean values from offshore waters (Sabine et al., 2004). Following other carbon system properties, the highest and most variable *RFs* were observed in the bottom waters during spring and summer. Photosynthetic uptake in the surface during upwelling months lowered *RFs* to between 10-12, with occasional values reaching below 10. Winter *RFs* for surface and bottom water samples converged around 12-14.

### 3.5 Spatial Variability

Full-year histograms of Ω$_{Ar}$ in kelp forest bottom waters revealed significant differences (Wilcoxon rank sum test, α=0.05) between the sites generally according to site depth and orientation (Fig. 9). There were no site-to-site significant differences in surface Ω$_{Ar}$ values, indicating greater homogeneity at the surface. Not only were there significant differences



in bottom water carbonate chemistry, but the bottom waters were spatially decoupled as well. Throughout the year, bottom water $\Omega_{Ar}$ from the protected shallower sites was far less correlated with the deeper and exposed sites relative to $\Omega_{Ar}$ of surface waters (Fig. 10). This decoupling was largely maintained during periods of both strong and weak upwelling (Fig. S8).

## 5  4. Discussion

This study demonstrates strong spatial and temporal variability in $CO_2$ system chemistry within a central California kelp forest using nearly 800 DIC and TA observations that ranged across six sites, multiple depths, and spanning a period of 14 months. This data set represents one of the largest multi-parameter $CO_2$ system data sets completed in a kelp forest and provides critical baseline data for detecting biogeochemical change in this system. We now consider the physical and
biological drivers of the observed variability and the implications of this work for understanding kelp forests in an era of global change.

### 4.1 Mechanisms of observed variability

Co-variation of water column structure and kelp canopy biomass provided evidence for the strong influence of regional scale upwelling processes on the seasonal cycle of the kelp canopy (Fig. 2a). Bottom water advection during
upwelling season introduces nutrients into the kelp forest, resulting in increased kelp growth and canopy biomass (Jackson, 1977). Unlike in southern California, where upwelling and canopy biomass are not tightly coupled, central California features very strong coupling between upwelling and kelp growth (Bell et al., 2015b). This strong co-variation provides support for asserting the role of both physical and biological drivers of kelp forest biogeochemistry.

The Protected Middle and Protected Inside sites may be decoupled from the other four sites through a reduction in
bottom water exchange. Bottom water velocities from July to December 2013, when data were available within the kelp forest on both the protected and exposed sides of the kelp forest, revealed larger velocity variability on the exposed side than on the protected side (Fig. 11). The exposed side featured stronger southeastern (negative alongshore) and onshore (negative cross shore) flow than did the protected side, which increased the exchange between offshore waters and the exposed side more so than on the protected side. Reduced bottom water velocities along the protected side are likely due to the shallow
depth and complex bottom structure, which restricted both along shore flow and cross shore exchange. We note that higher frequency water sampling may have captured the biogeochemical signature of occasional bottom water advection into the shallower sites that was missed through weekly sampling alone.

In light of all the evidence for the role of physical processes in shaping the biogeochemistry of the kelp forest, we emphasize that photosynthesis still played an important role in shaping the carbon system variability in the surface waters.
Surface water $pCO_2$ remained undersaturated with respect to the atmosphere starting at the beginning of upwelling season and persisted throughout the summer months (Fig. 5). The persistent undersaturation of surface waters is strong evidence of





photosynthesis' role in contributing to biogeochemical variability in spring and summer months, particularly in the surface waters where the bulk of kelp biomass resides. Although we are unable to quantitatively partition the photosynthetic activity between kelp and phytoplankton in the kelp forest, we note that surface $pCO_2$ was undersaturated as early as February 2014 (Fig. 5b), when kelp canopy biomass was near minimum values (Fig. 2a). We hypothesize that early season $CO_2$ uptake was

controlled by the phytoplankton community in the absence of kelp. Once the kelp canopy was established several months later, the kelp played an important biogeochemical role both through direct photosynthetic uptake, as well as reducing water velocity within the kelp forest (Rosman et al., 2007), which allowed greater biogeochemical modification of surface water chemistry through phytoplankton and kelp metabolism.

This hypothesis is testable using offshore sampling sites outside the influence of the kelp forest to better quantify

the phytoplankton and kelp community contributions to the observed biogeochemical variability. However, our offshore sites were only tens of meters outside of the kelp. Therefore, it is not surprising that we are unable to distinguish surface waters from within the kelp forest from those just outside the kelp. Future sampling efforts would benefit from utilizing offshore sites further from the influence of the kelp, as well as to utilize phytoplankton-specific tracers, such as silica (Jackson, 1977), in order to better partition the biogeochemical contributions from various functional groups within the kelp forest

community.

### 4.2 Implications for understanding kelp forest ecosystems in an era of global change

Our data highlight the necessity of long-term, spatially expansive sampling for gaining insights into the patterns and controls on kelp forest biogeochemistry. Kelp forests are spatially and temporally dynamic environments generally found along upwelling margins, so any assessment of their biogeochemical variability must account for the variability in their

seasonal-to-interannual physical (Bograd et al., 2009; Checkley and Barth, 2009) and biological controls (Bell et al., 2015b). We have demonstrated how variations in these physical and biological controls in a central California kelp forest create biogeochemically heterogeneous environments. This creates the opportunity to do localized "space-for-time" experiments to understand benthic organismal and community response to high $CO_2$ seawater. In contrast to larger space-for-time experiments (Hofmann et al., 2014), a single kelp forest may be sufficient for replicating the geochemical gradients expected

between current conditions and future with OA conditions. These localized space-for-time experiments could be conducted in small environments (hundreds of meters, not hundreds of kilometers) with near identical biological assemblages and for a fraction of the cost and effort. Combining transplant experiments with observational work could leverage the biogeochemical differences between areas of a kelp forest to yield important insights into the adaptive potential of benthic community members under the combined stresses of global change.

Aqueous $CO_2$ variability typically does not occur in isolation of other biogeochemical and environmental changes. Although we do not have the $O_2$ measurements to accompany our carbon system time series, co-variation of $CO_2$ and $O_2$ has been well documented in California's coastal ecosystems (Booth et al., 2012; Frieder et al., 2012; Takeshita et al., 2015) and frequent episodic hypoxia has been documented near our study site (Booth et al., 2012). While a large component of this



variation is naturally occurring along upwelling margins, low $O_2$/high $CO_2$ events are predicted to increase in frequency in the future due to oxygen minimum zone expansion (Bograd et al., 2008; Booth et al., 2014; Stramma et al., 2010) and increases in upwelling favorable winds (Bakun, 1990; Sydeman et al., 2014). Understanding the co-variation, or lack thereof, between critical environmental parameters (e.g., temperature, $CO_2$ system variables, and $O_2$) through long-term spatially

expansive measurements is necessary to define the range and timescales of environmental conditions experienced by kelp forest inhabitants. Quantifying these environmental conditions can inform the design of more realistic laboratory experiments, featuring proper ranges and timescales of biogeochemical and thermal variability as opposed to chemostatic conditions (Reum et al., 2015). More realistic experimental conditions will yield additional insights into organismal and community response to climate change and OA.

Our data demonstrate that, despite the strong influence of physical processes, primary production can alter local biogeochemistry. Understanding the role of foundation species, such as giant kelp, in creating biogeochemical refugia both through metabolic activity and alteration of the hydrodynamic regime within the kelp forest via increased residence time (Rosman et al., 2007) warrants further attention in an increasingly acidic ocean, as they present potential to mitigate some stress effects of low $O_2$, high $CO_2$ water. The combination of physical and biological processes may create natural refugia for

certain sensitive organisms or, conversely, create particularly stressful conditions for others. Organisms that can confine themselves to the upper water column may be able to use the locally created biogeochemical refuge to avoid high acidity, whereas organisms that must use the entire water column will experience large ranges in carbonate chemistry (pH, $pCO_2$, and $\Omega_{Ar}$) that will only grow larger with continued acidification.

In order to effectively manage critical coastal ecosystems in the face of climate change and OA, resource managers

require both monitoring data and a process-based understanding of biogeochemical variability in order to identify changing environmental conditions and forecast ecosystem responses in kelp forests (Boehm et al., 2015). Such understanding is currently lacking. Along the US west coast, states currently use decades-old water quality criteria for assessing pH, including that the pH should not drop below 6.5 and/or that it should not deviate more than 0.2 units from natural conditions (Weisberg et al., 2016). Yet we have demonstrated changes up to 0.5 pH units within a single kelp forest in just 15 meters depth.

Additional observational studies may prove useful in helping us refine our understanding of coastal water quality away from static indicators and towards a more dynamic understanding of the ranges and controls on coastal water quality in order to better differentiate natural and anthropogenic effects, as has been demonstrated for dissolved oxygen (Booth et al., 2014). We recommend that supervising agencies revise existing regulations to incorporate this understanding of natural variability in carbonate chemistry into their regulatory frameworks, as well as to expand the suite of carbonate chemistry water quality

variables beyond pH alone (Weisberg et al., 2016). Developing stronger biological criteria for carbonate chemistry ranges and variability deemed necessary to preserve critical marine living resources would help align scientific and management priorities by focusing research efforts on quantifying these acceptable ranges of conditions (Chan et al., 2016; Weisberg et al., 2016).





Comprehensive, spatially expansive monitoring is fundamental to characterizing the ranges and timescales of carbonate system variability in highly productive coastal ecosystems, such as kelp forests. Maintaining monitoring networks along coastal ecosystems is a crucial step to developing an understanding of natural vs. anthropogenic influences in the coastal zone in support management and regulatory efforts (Strong et al., 2014). Recent advances in sensor technology, such

as the SeapHOx (Bresnahan et al., 2014), may make monitoring far less laborious than the data collection efforts in this study. However, we strongly recommend that monitoring programs incorporate multiple carbonate system parameters (TA, DIC, pH, $pCO_2$) so as to fully constrain the carbonate system (McLaughlin et al., 2015). Coupling SeapHOx sensors to autonomous $pCO_2$ (as discussed in Martz et al. 2015) or newly developed autonomous TA sensors (Spaulding et al., 2014) may accomplish this task. Integrating observations from individual sites along the California coast into larger coordinated

networks, such as the California-Current Acidification Network, will provide a more complete perspective on the spatial and temporal variability of the aqueous $CO_2$ system and its controls (Chan et al., 2016; McLaughlin et al., 2015).

Long-term observational data sets are also crucial for improving our process-based understanding of coastal ocean biogeochemistry. Data from studies such as these can be used to parameterize the process-based hydrodynamic-biogeochemical models that are needed to predict coastal ecosystem responses to climate change and OA (Boehm et al.,

2015; McLaughlin et al., 2015). These models can help identify climate change and/or OA hotspots along coastal ecosystems in the future and help marine living resource managers with scenario planning so that they can direct resources accordingly (Strong et al., 2014). Similarly, marine resource managers must consider the biogeochemical implications of any management activities, such as harvesting, which alter kelp canopy density. We have shown considerable $CO_2$ uptake at our study site during periods of high kelp canopy cover. While we were not able to directly attribute the $CO_2$ uptake to

*Macrocystis pyrifera*, the direct and indirect effects of kelp canopy are likely to increase vertical gradients in water chemistry and therefore activities that decrease canopy density are likely to chemically homogenize the water column.

Kelp forest ecology has long been a focal point for marine ecologists. This study complements the long history of kelp forest community (Dayton, 1985; Graham et al., 2007) and disturbance ecology (Edwards, 2004; Edwards and Estes, 2006) by highlighting the dynamic biogeochemistry of kelp forests over an annual cycle. The few existing biogeochemical

studies of *Macrocystis pyrifera*-dominated kelp forest along the California coast have thus far occurred in southern California (Frieder et al., 2012; Kapsenberg and Hofmann, 2016; Takeshita et al., 2015). This study expands on the southern California studies by adding the first fully resolving $CO_2$ system chemistry study in a central California kelp forest where seasonality is relatively stronger and kelp cover is more variable (Bell et al., 2015b; Checkley and Barth, 2009; Reed et al., 2011).

We hope that the large data set presented here motivates the inclusion of biogeochemistry as a new approach to understand kelp forest ecosystem function and the feedbacks between biology, chemistry, and physics in these dynamic systems. We note that this study would not have been possible without a large and concerted field effort to establish a spatially expansive sampling program that ranged over gradients in kelp density and wave exposure. Yet this sampling program was necessary to generate insights into the relevant scales of kelp forest biogeochemical variability presented in this



study, variability that could not be discerned from a single sampling point or sensor alone. We advocate for monitoring efforts to extend beyond single site measurements within a kelp forest. Doing so will enhance our understanding of the biological and physical mechanisms giving rise to the observed biogeochemical variability; information crucial to accurately predicting the response of kelp forest ecosystems to global change.

**Supplement and Data Availability**

Supporting figures and tables can be found in the supplement to this study. Data presented in this study are also available in the supplement.

**Conflict of Interest**

The authors declare no conflict of interest

**Acknowledgements**

Heidi Hirsh, and Hans DeJong provided laboratory assistance. Kristin Elsmore, Ana Guerra, Jesse Lafian, Diana LaScala-Gruenewald, Lupita Ruiz-Jones, Drew Simon, and Tim White provided field assistance. Fio Micheli provided laboratory facilities to use in support of this project. Bill Gilly provided the use of the CTD in this study. Conversations with Andreas Andersson, Kevin Arrigo, Tyler Cyronak, Stephen Monismith, and Yui Takeshita improved this study. The School of Earth,
Energy, and Environmental Sciences and the Marine Life Observatory of Stanford University supported this project. DAK and PRL were supported by NSF Graduate Research Fellowships (DGE-114747). KJN was supported by the Marine Life Observatory of Hopkins Marine Station. SYL was supported by the Marine Life Observatory of Hopkins Marine Station and the NSF Ocean Acidification Program (OCE-1416934). TWB was supported through the NASA Earth and Space Science Fellowship. The satellite-derived canopy biomass estimates were supported by NSF through its support of the Santa Barbara
Coastal LTER (OCE-0620276 and OCE-1232779). This research was conducted under Monterey Bay National Marine Sanctuary permit number MBNMS-2010-002-A2.

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





| Site | Latitude (°N) | Longitude (°W) | Depth (m) | Distance from shore (m) |
|---|---|---|---|---|
| Protected Offshore | 36.621983 | 121.900883 | 16 | 250 |
| Protected Middle | 36.6216 | 121.90176 | 9 | 160 |
| Protected Inside | 36.62132 | 121.90195 | 7.5 | 130 |
| KFA | 36.62363 | 121.90473 | 14 | 270 |
| Exposed Offshore | 36.62316 | 121.90503 | 13.5 | 200 |
| Exposed Middle | 36.62093889 | 121.9003806 | 11 | 160 |

**Table 1: Study site characteristics. Distance from shore measured from the closest perpendicular location.**





| Variable | Top/Bottom | Minimum | Mean | Maximum |
|---|---|---|---|---|
| DIC ($\mu$mol kg$^{-1}$) | Top | 1853 | 2023 | 2110 |
| | Bottom | 1958 | 2093 | 2225 |
| TA ($\mu$mol kg$^{-1}$) | Top | 2213 | 2243 | 2277 |
| | Bottom | 2199 | 2249 | 2337 |
| pH | Top | 7.92 | 8.08 | 8.33 |
| | Bottom | 7.70 | 7.96 | 8.21 |
| pCO$_2$ ($\mu$atm) | Top | 172 | 364 | 543 |
| | Bottom | 249 | 508 | 952 |
| $\Omega_{Ar}$ | Top | 1.63 | 2.43 | 3.91 |
| | Bottom | 0.94 | 1.82 | 3.23 |

**Table 2: Carbon system variables summary statistics aggregating all six sampling sites**



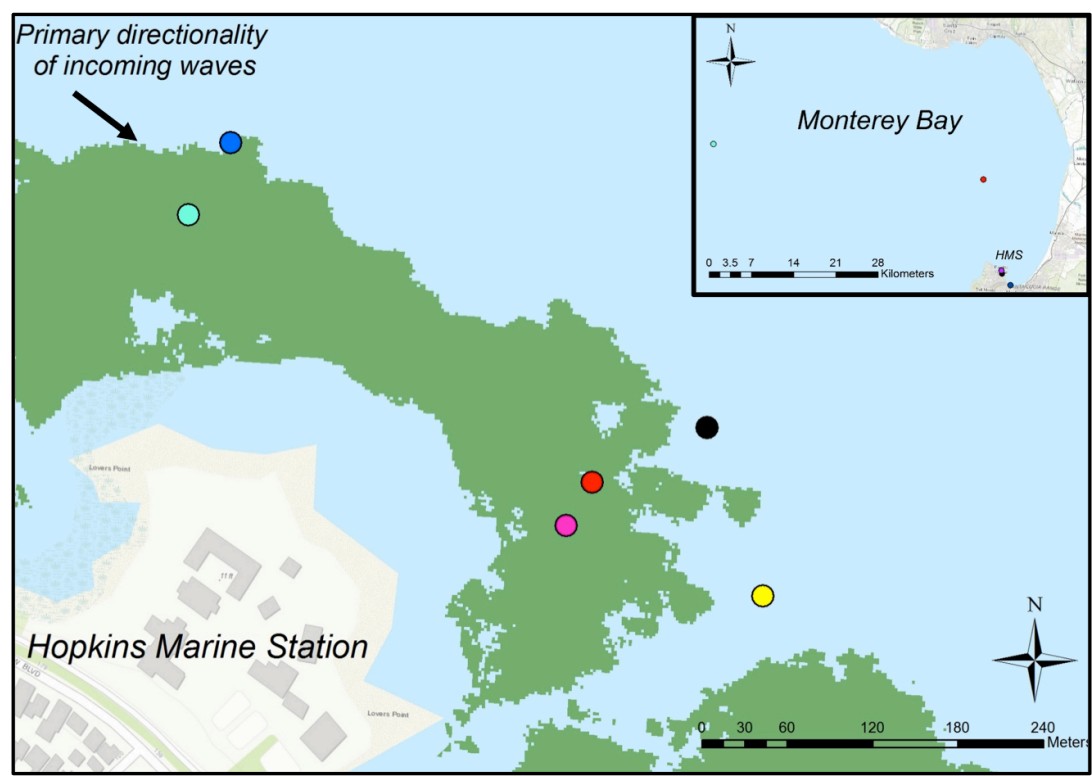

**Figure 1: Study site: kelp forest offshore of Hopkins Marine Station. Green shows historical average kelp canopy extent. Site names are as follows: Protected Offshore (Pro Off; black), Protected Middle (Pro Mid; red), Protected Inside (Pro Inn; magenta), Kelp Forest Array (KFA; yellow), Exposed Offshore (Exp Off; blue), and Exposed Middle (Exp Mid; cyan). Inset shows Monterey Bay with color dots corresponding to the data buoys shown in Fig. S1.**





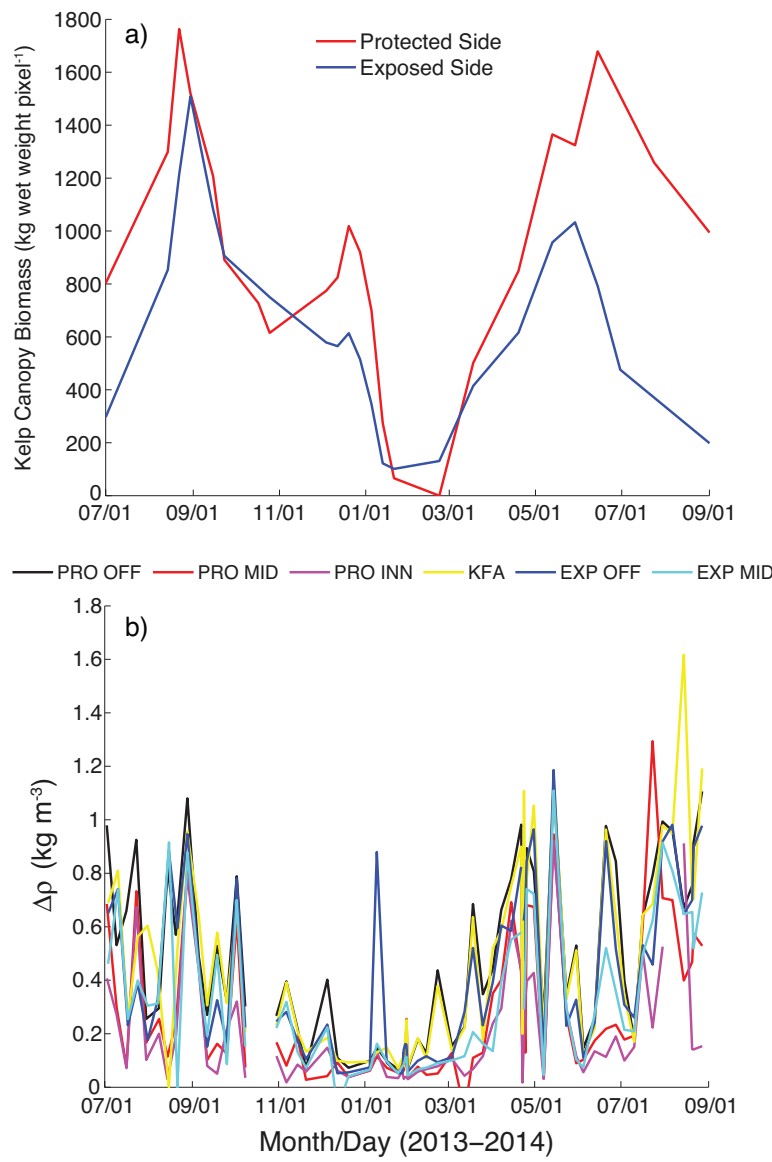

**Figure 2:** Time series of a) satellite-derived kelp canopy biomass per 30m x 30m satellite pixel for the protected and exposed sides, and b) $\Delta\rho$ for all six sites.





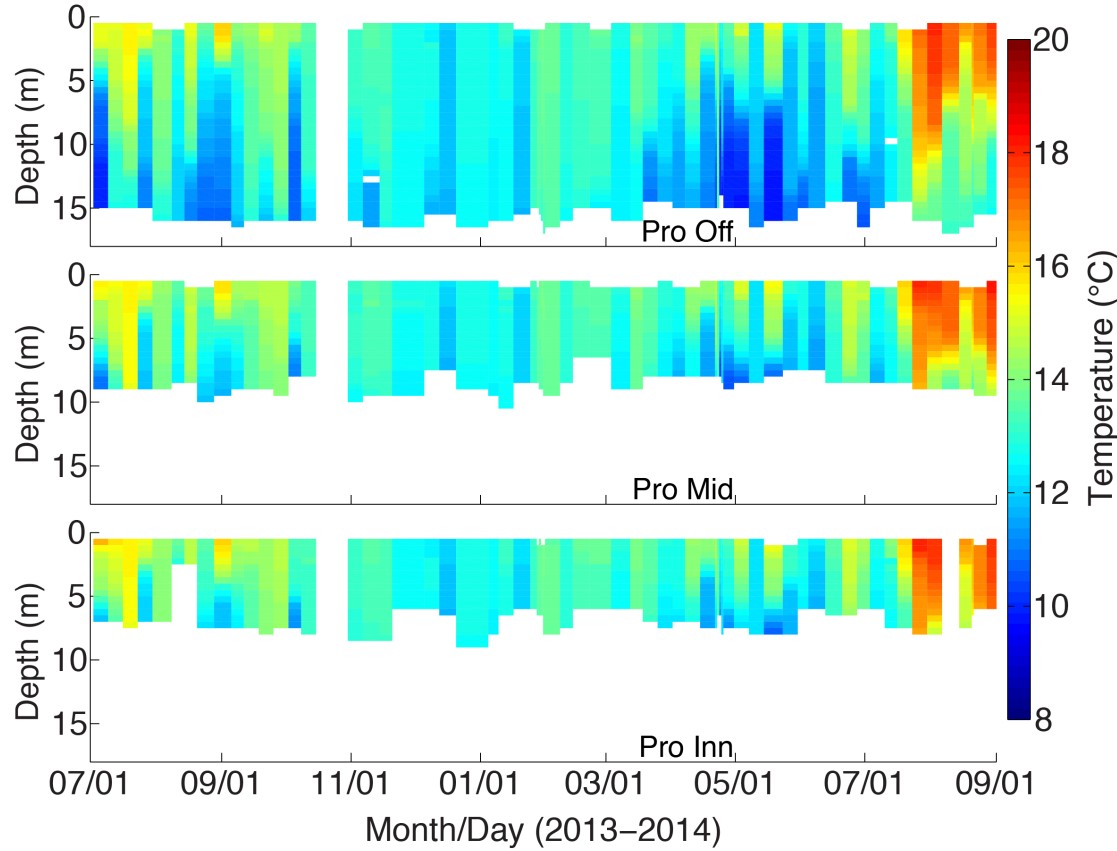

**Figure 3:** Depth-resolved temperature profiles along the protected transect throughout the 14-month time series. White spaces, including October 2013 and August 2014, are missing data due to the CTD not recording or poor data removed during quality control.





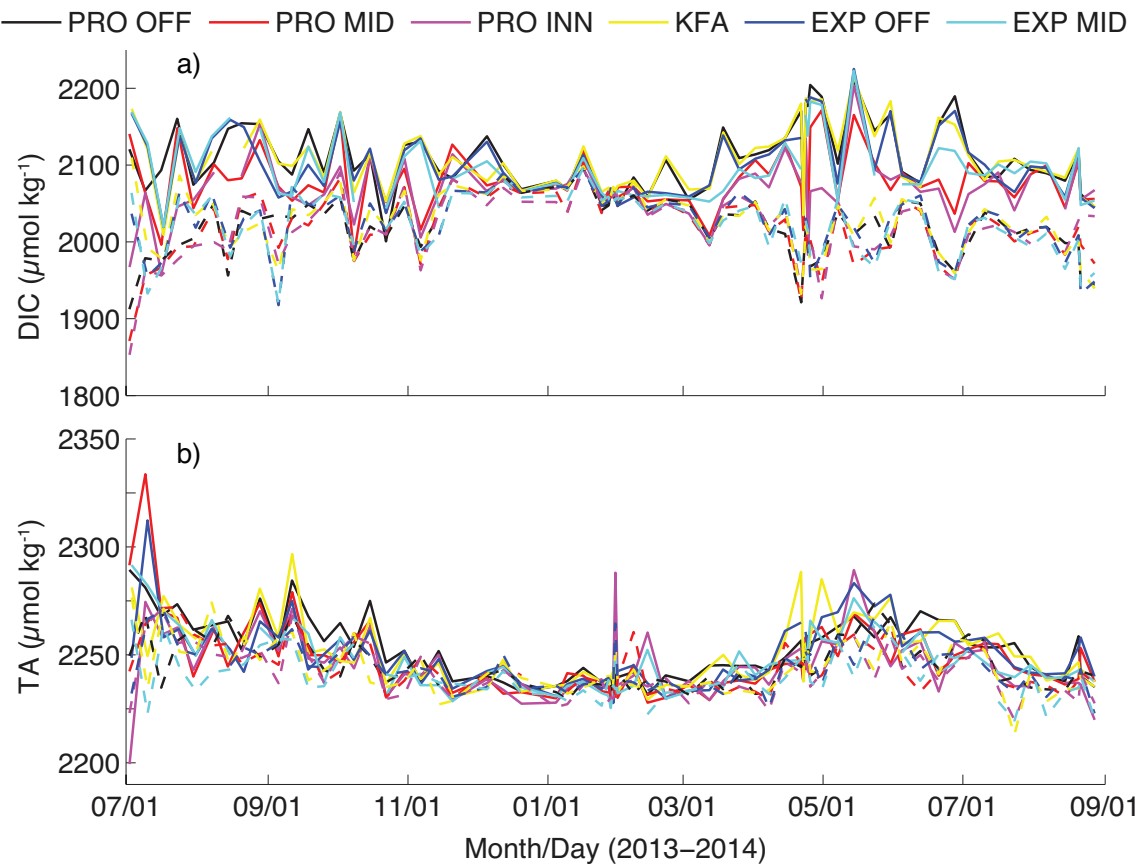

Figure 4: Time series of a) DIC and b) TA. Solid lines are bottom samples and dashed lines are surface samples.



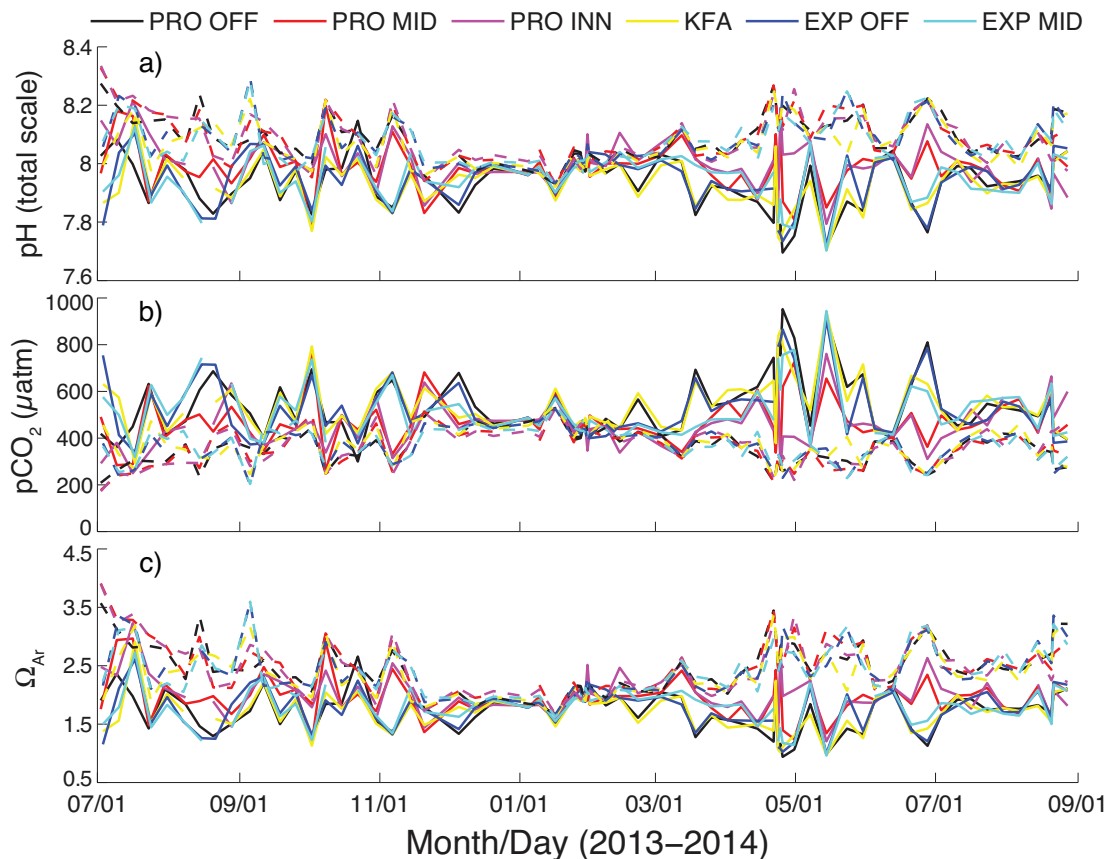

Figure 5: Time series of a) pH, b) pCO₂, and c) Ω_Ar calculated from TA and DIC measurements. Solid lines are bottom samples and dashed lines are surface samples.



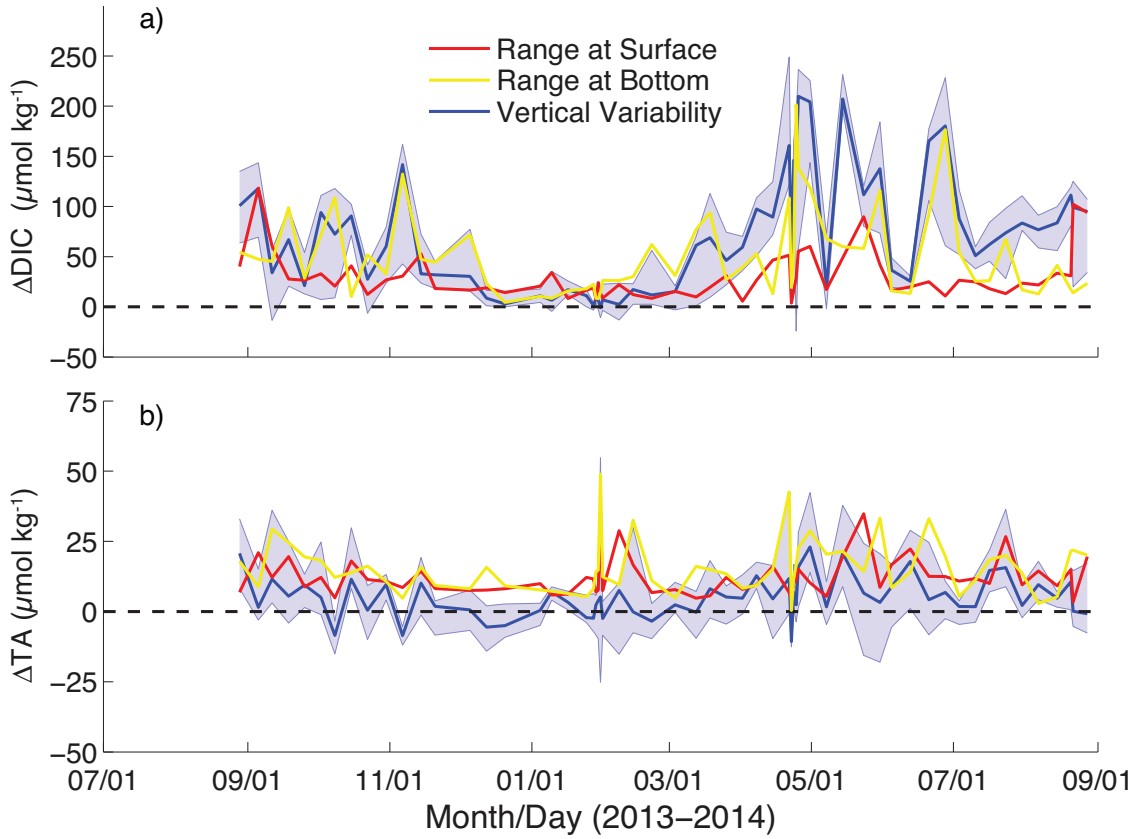

**Figure 6: Variability throughout the sampling period for a) DIC and b) TA. The range of surface values across all six sites is shown in red and the range of bottom values across all six sites is shown in yellow. The blue line is the median surface-to-bottom difference in DIC or TA during a given sampling day and the blue shaded region represents the maximum and minimum vertical gradients observed on each day. Note the differences in scale between a) and b).**





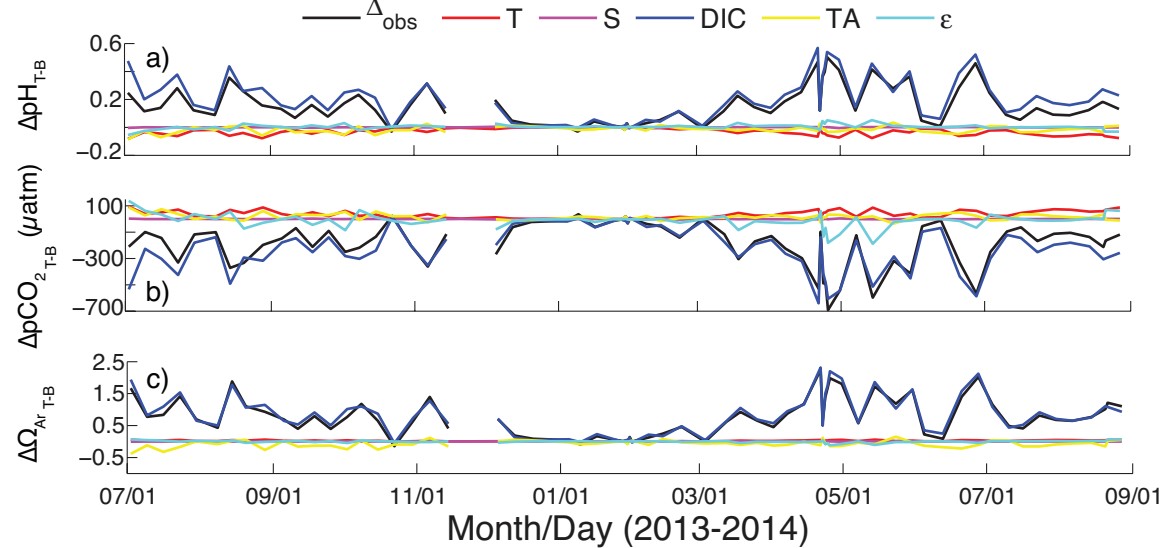

**Figure 7: Observed (black) surface-to-bottom differences in a) pH, b) $pCO_2$, and c) $\Omega_{Ar}$ at the Protected Offshore site along with the calculated contributions of temperature (red), salinity (magenta), DIC (blue), and TA (yellow) to the observed carbon system gradients. The residual difference between the observed and calculated contributions is shown in cyan.**




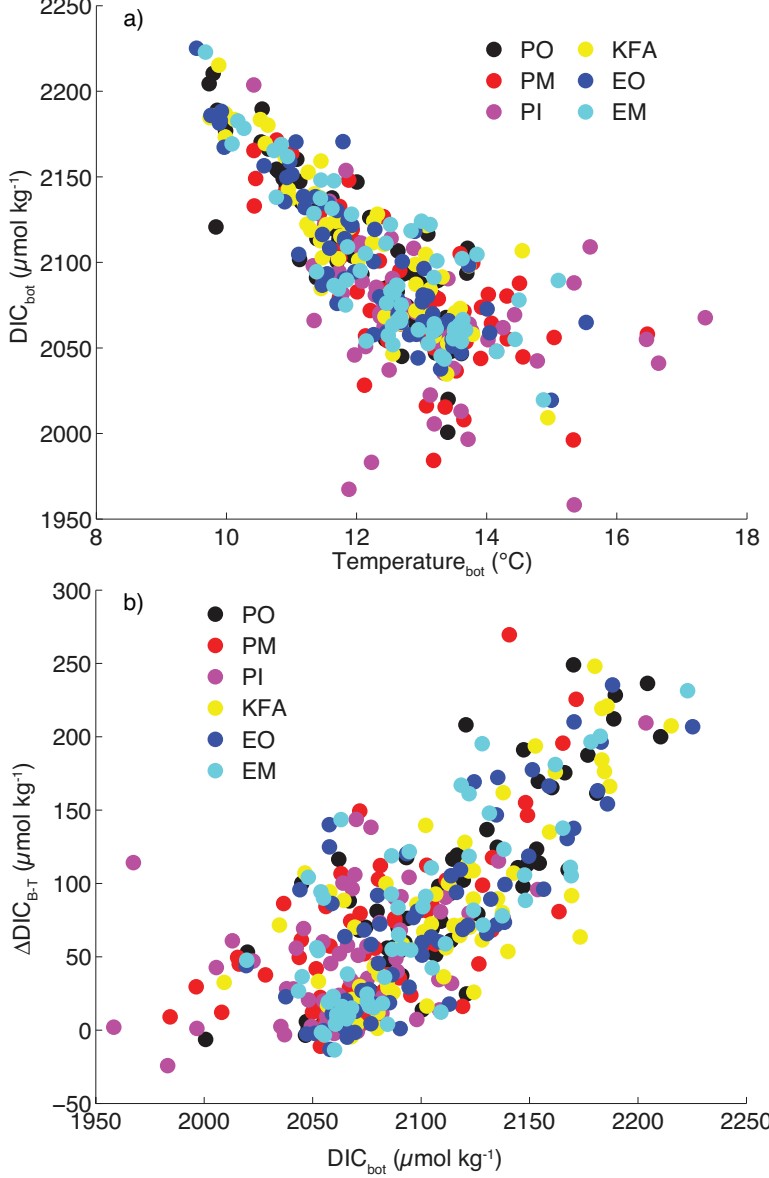

**Figure 8: a) Bottom DIC-bottom temperature relationships for all six sites and b) bottom-to-surface DIC gradients as a function of bottom DIC concentrations.**




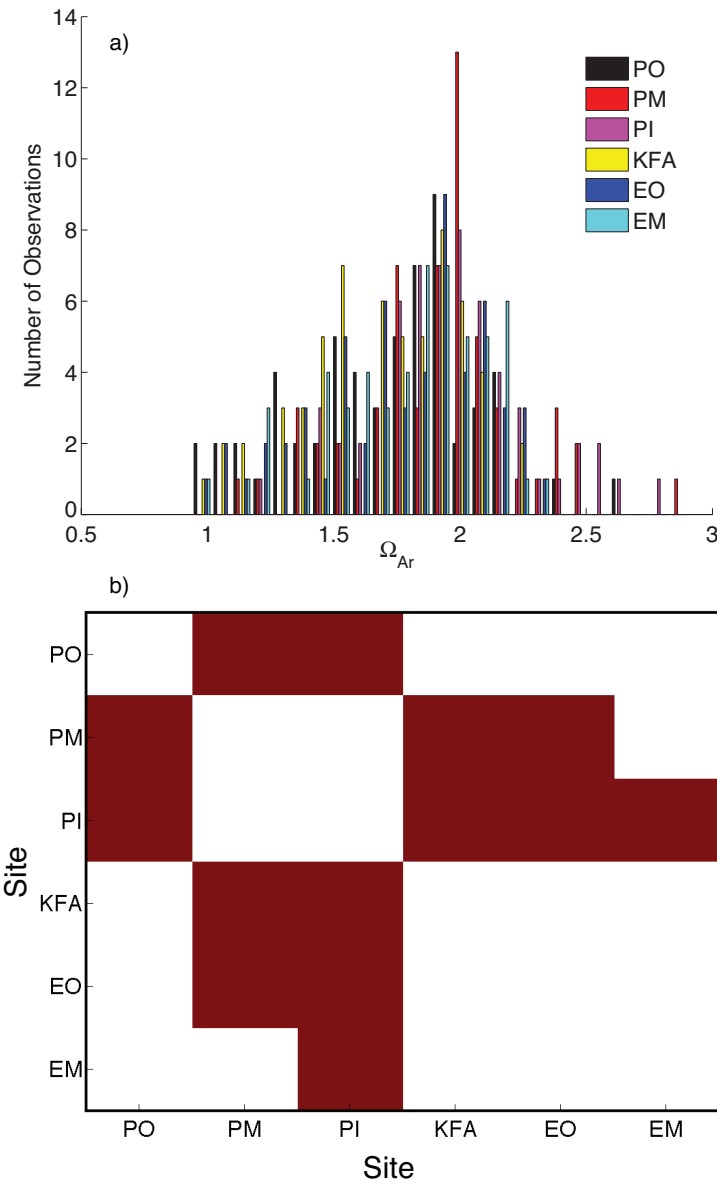

Figure 9: a) Histogram of bottom $\Omega_{Ar}$ values (excluding July and August 2013, when samples were not all collected on same day) and b) results of site-to-site bottom $\Omega_{Ar}$ Wilcoxon rank sum test. Abbreviations on the sides refer to the six sites in sequence shown in the legend in a). Colored patches indicate significantly different (p<0.05) bottom $\Omega_{ar}$ between the two sites.





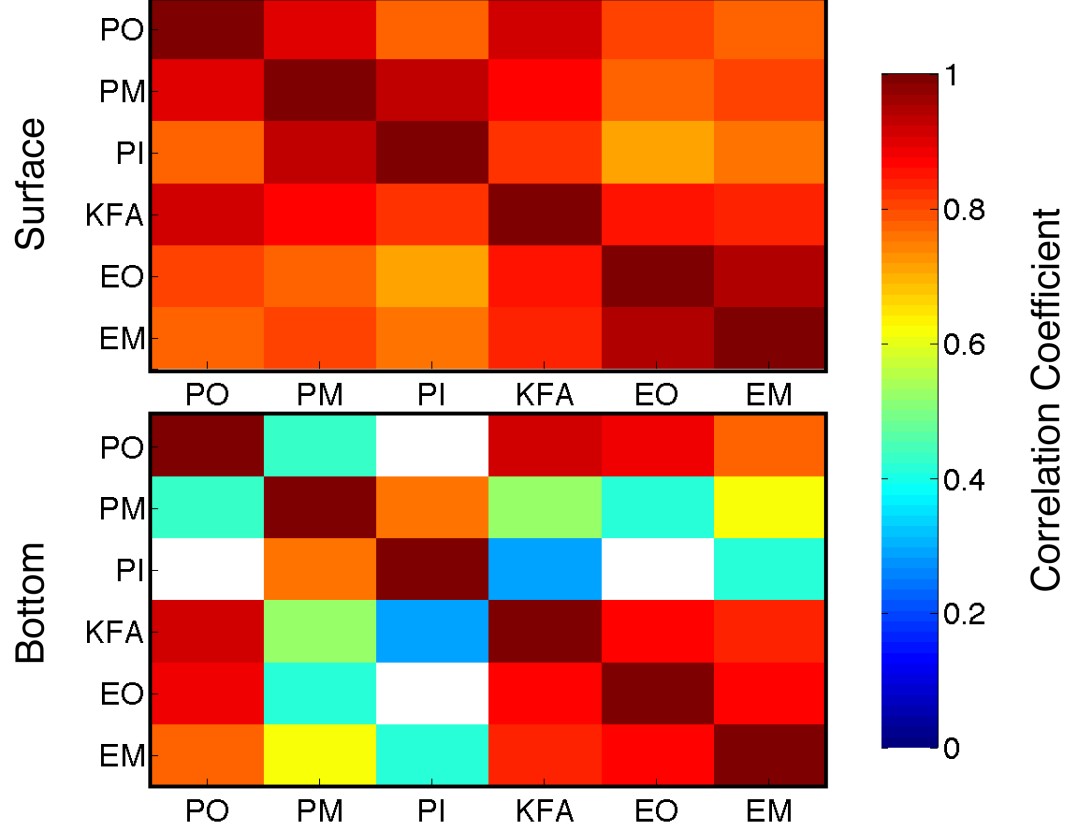

**Figure 10: Correlation heat maps of $\Omega_{Ar}$ for the full year. Top panel shows correlations between surface sites and the bottom panels shows correlations between bottom sites. Site names are abbreviated on both panels and correspond to the order presented in the legend of Figure 9a. Blank spaces show statistically insignificant (p>0.05) correlations.**



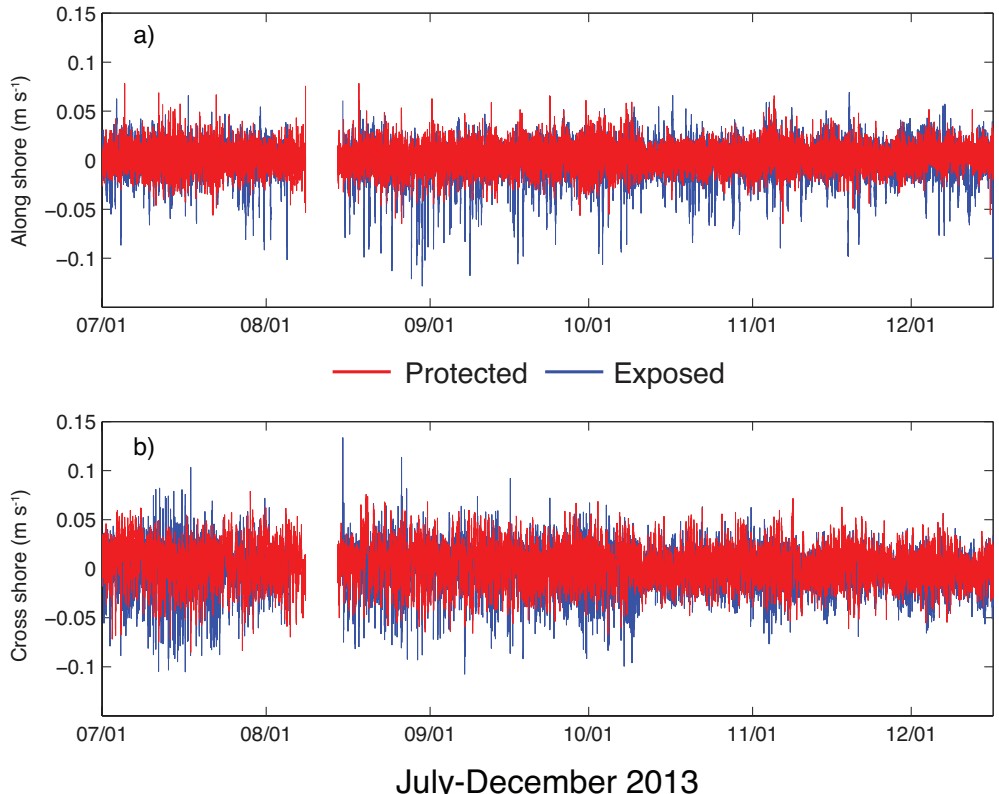

**Figure 11: Depth-averaged (0-5 mab) bottom a) along shore and b) cross shore velocities from the Protected Mid and Exposed Mid ADCPs collected from July-December 2013. Positive along shore is 40° counter clockwise from north at Protected Mid and 13° counter clockwise from north at Exposed Mid. Positive cross shore velocities are 90° clockwise from positive along shore velocities (offshore).**

