# Peer review of "A year in the life of a central California kelp forest: physical and biological insights into biogeochemical variability"

_Biogeosciences, 2016_

## Referee Comment (RC1) · Anonymous Referee #1 · 4 Oct 2016

General Comments:

This study presents 1+ year time-series data of weekly samples of carbonate chemistry across a small spatial scale of a kelp forest covering two summer seasons. The data include surface and bottom samples in exposed and protected sites and from inside to outside the kelp forest. The data are of extremely high quality. The paper is well written, articulate, and has logical organization with nice transitions. While carbonate chemistry time series papers are increasing in number, this paper contributes novel and valuable data on small scale spatial variability (depth and spatial). In support of publishing this paper, I consider my comments as minor revisions which would improve the scientific quality from 'good' to 'excellent'.

Specific Comments:

I have three specific comments, two with regard to the spatial variability. First, bottom water sampled by site is confounded by depth, which is not explicitly addressed. The spatial variation of bottom water could just be an artifact of the stratification of the water column within which the kelp forest sits (deeper waters have more DIC, so therefore bottom waters of deeper sites will have lower DIC values than bottom waters of shallower sites). The potential depth dependency of the observed dynamics (and conclusions) should be addressed and contextualized with the aims of the study (and the sampling design of surface and bottom waters, which was not explained). The data are valuable in terms of understanding the variation of what, for instance, a benthic kelp forest inhabitant might experience, but then that perspective should be included (Introduction and Discussion).

Second, the most valuable portions of this study are the depth gradient (well developed and presented) and the spatial gradient of the time-series (from inside to outside a kelp forest, exposed to protected). The presentation of the latter (Section 3.5) is extremely short and the figures comprise mostly of statistical numbers and not meaningful observations. The authors do themselves a disservice by not highlighting this aspect of the study more in depth. Figures 9 and 10 do not contribute anything that could not be shown in a table (Fig. 9b, 8, and S8 display duplicate data in every plot). Fig. 9a could be interesting if shown as a line graph (bar graph is too cluttering) but I don't think it's necessary in the first place. Instead, I was expecting a figure showing the gradient in carbonate chemistry from inside to outside the kelp forest at the two contrasting sites (protected, exposed). How does these gradient change by season? It looks like the largest spatial differences occur between the exposed vs. protected site and not within the inside vs. outside (I suspect that differences between inside and outside kelp forest will only be apparent with higher frequency sampling). The statistics show this, but the figure space would be better used by using the real data (e.g. select parts of the time-series, moving averages, etc.).

Lastly, the Discussion is largely devoted to the value of time-series, this could probably be condensed. As an edition, the results should be discussed in terms in the context of other studies of kelp forest or coastal variability in general (some were mentioned in the Introduction). Do these data fall within the range of biochemical observations made previously in other kelp forests?

Other and Technical Comments:

Shorten the LPJPSMR acronym

2.1 L23: of kelp of the kelp

2.4 L6: provide reason or reference for phosphate assumption

2.4 L8: pHT is defined but not used in subsequent reporting of pH values in the Results.

2.4 L11: double ))

For all time series figures: simplify x-axis date labels. Adding 01 as the day is not necessary and adds clutter. I recommend to simply label months as 1-12.

3.2 L23: "causing water column temperature differences of up to 4°C" add across what range of depths

3.3 L 16-17: "Surface DIC concentrations were generally much more spatially homogeneous than bottom water DIC concentrations." Could just be function of depth.

3.3 L25: has should be had

3.3 L27: regarding pCO2 undersaturation, add "with respect to the atmosphere" if that is what you are measuring saturation against.

Table 2: Since bottom depth differs across sites (7.5-16 m) and there are obvious depth effects, add the depth in m in () following the listed value in Table 2.

3.4 L21: "drove large variations in the ability to buffer against ongoing ocean acidification". Ocean acidification is not detectable across such a short time-series. Reword to

simply say, "drove large variations in the Revelle Factor".

3.5: Why was aragonite used here (and not TA or DIC)?

4.1 L28: Regarding this paragraph, it would be nice if you could find a reference showing the seasonality of phytoplankton blooms of this site. I imagine it is offset from the kelp forest growing season.

4.2 L22 - State the actual findings. The largest source of variation seems to be the protected vs. unprotected sites, which is actually a function of the oceanographic features, not a function of the biology of the kelp forest. The biological control is in this study is the depth gradient (where primary production takes up DIC at the surface).

Pg. 11 L9: inconsistent use of OA vs. ocean acidification. I recommend to not use the acronym at all.

Pg 11, L10: other studies have shown this previously also: pH sensor-based studies in coastal environments but also cruise data for offshore regions. Cite references in support of this conclusion.

Pg. 11 L13: The ocean is not acidic, acidifying and acidic are different.

Pg. 11 L16: Same as previous comment. It makes more sense to use 'low pH' instead of 'high acidity'

---

## Referee Comment (RC2) · Anonymous Referee #2 · 10 Oct 2016

This manuscript nicely describes spatial and temporal variation in carbon system variables in a kelp forest in Central California. The data presented are the first to report high frequency measurements of carbon system variables made at small spatial scales across an entire annual cycle within a kelp forest. The data reveal substantial depth-dependent, spatial, and seasonal differences. The authors suggest mechanisms that could be responsible for creating the observed variation. Sampling design, sample collection, and analytical methods all appear appropriate to the research questions posed. The organization of the paper is logical, the writing is clear, and the graphics are appropriate and informative.

Below I offer specific comments intended to strengthen the manuscript.

Page 2, line 8: more clearly stated as "Calcification and dissolution of kelp-associated organisms, especially shelled invertebrates, can modify water chemistry..."

Page 2, line 16: better stated as " Despite the recognized importance of the kelp forest..."

Page 2, line 16: I searched but did not find reference to kelp as an "ecosystem architect". The earlier use of the term "foundation species" is more consistent with the ecological literature.

Page 2, lines 25-27: Important points are made here. It would be helpful to clearly return to these in the discussion section.

Section 2.4 Satellite derived estimates: Estimating kelp canopy biomass is notoriously difficult. The authors have done a good job estimating relative changes in biomass over time but I found no indication in the text or figures to indicate error in this estimate. The inclusion of error estimates would be helpful.

Section 3.4: Carbon systems variables differ between surface and bottom, consistent with the intrusion of CO2 enriched water at bottom and photosynthetic activity at the surface. Here or in the discussion it could be helpful to mention that the observed surface-to-bottom variation suggests that benthic calcifiers appear neither to be influencing TA nor do they appear to be benefitting from the effects of photosynthesis on water chemistry, which seem to be confined to surface waters. Moreover, understory seaweeds, which can achieve substantial biomass in kelp forests, don't appear to affect water chemistry appreciably (tho this was not tested). A fuller discussion of these considerations could be helpful.

Section 4.2: The discussion of "space-for-time substitutions" is reasonable, but in my opinion is less compelling than other arguments that can be made concerning kelp forest ecosystems in an era of global change. I'd encourage the authors to open the discussion with the most compelling inferences that can be drawn from their data.

Page 11, lines 10-18: The discussion of refugia could be refined. Assuming that photosynthesis within the canopy modulates stress due to high CO2/low pH, it's difficult to think of very many organisms (especially calcifying organisms) that can take advantage of this. These are likely to be limited to epibionts on kelp blades and perhaps a few canopy-associated fish species. A much larger number of calcifying taxa are associated with the benthos, where water conditions are likely to be less conducive to calcification and growth when omega is low. Consequently, the potential refugium created by the canopy is spatially unassociated with the bulk of benthic species. Moreover, the persistence of refugia in such a dynamic system is questionable.

Page 11, lines 25-34: Comments about water quality criteria are reasonable: for instance, it's important to point out that the variability observed in this study exceeds that of water quality criteria now in existence. However, the paragraph doesn't seem particularly nuanced, beyond references to Boehm et al, Weisberg et al, and Chan et al. I encourage the authors to more fully consider the implications of their data for water quality criteria.

Page 11, line 28: replace "supervising agencies" with "regulatory agencies".

Page 12, line 8: distinguish between "fully constrained" and "over-constrained" with respect to carbon system variables.

Page 12, line 18: replace "harvesting" with "kelp harvesting".

Page 12, line 20: should read "effects of the kelp canopy".

Page 12, line 22: replace "chemically homogenize" with "reduce gradients in".

---

## Author Comment (AC1) · 21 Nov 2016

**General Comments:**
**This study presents 1+ year time-series data of weekly samples of carbonate chemistry across a small spatial scale of a kelp forest covering two summer seasons. The data include surface and bottom samples in exposed and protected sites and from inside to outside the kelp forest. The data are of extremely high quality. The paper is well written, articulate, and has logical organization with nice transitions. While carbonate chemistry time series papers are increasing in number, this paper contributes novel and valuable data on small scale spatial variability (depth and spatial). In support of publishing this paper, I consider my comments as minor revisions which would improve the scientific quality from 'good' to 'excellent'.**

*Thank you for your thoughtful comments and careful review of this paper. We address your specific comments below.*

**Specific Comments:**
**I have three specific comments, two with regard to the spatial variability. First, bottom water sampled by site is confounded by depth, which is not explicitly addressed. The spatial variation of bottom water could just be an artifact of the stratification of the water column within which the kelp forest sits (deeper waters have more DIC, so therefore bottom waters of deeper sites will have lower DIC values than bottom waters of shallower sites). The potential depth dependency of the observed dynamics (and conclusions) should be addressed and contextualized with the aims of the study (and the sampling design of surface and bottom waters, which was not explained). The data are valuable in terms of understanding the variation of what, for instance, a benthic kelp forest inhabitant might experience, but then that perspective should be included (Introduction and Discussion).**
*We agree that the bottom water sample variability is confounded by depth variability. We will make this point more explicit in a revised manuscript. However, we still see robust differences between bottom water samples at similar depths, which hint at other localized processes controlling biogeochemical variability, something we already discuss in the manuscript.*

**Second, the most valuable portions of this study are the depth gradient (well developed and presented) and the spatial gradient of the time-series (from inside to outside a kelp forest, exposed to protected). The presentation of the latter (Section 3.5) is extremely short and the figures comprise mostly of statistical numbers and not meaningful observations. The authors do themselves a disservice by not highlighting this aspect of the study more in depth. Figures 9 and 10 do not contribute anything that could not be shown in a table (Fig. 9b, 8, and S8 display duplicate data in every plot). Fig. 9a could be interesting if shown as a line graph (bar graph is too cluttering) but I don't think it's necessary in the first place. Instead, I was expecting a figure showing the gradient in carbonate chemistry from inside to outside the kelp forest at the two contrasting sites (protected, exposed). How does these gradient change by season? It looks like the largest spatial differences occur between the exposed vs. protected site and not within the inside vs. outside (I suspect that differences between inside and outside kelp forest will only be apparent with higher frequency sampling). The statistics show this, but the figure space would be better used by using the real data (e.g. select parts of the time-series, moving averages, etc.).**
*We will re-organize Figure 9 to move the current Fig. 9B to Fig. 9A. This will help highlight the clustering of the Pro Inn and Pro Mid sites relative to the other four sites. We will move the current Fig. 9A to 9B and modify the figure to show the histogram of aragonite saturation state observations for: Pro Inn + Pro Mid and the other four sites (represented as two*

*clusters). This will help better illustrate the spatial differences between Pro Inn and Pro Mid relative to the four deeper sites. We will augment the text to describe this clustering in more detail. We will also provide additional text to better clarify the unique and complementary contributions of the new Fig. 9 and the current Fig. 10 (likely will remain unchanged). Specifically, Fig. 9 shows differences in average conditions, whereas Fig. 10 shows differences in co-variation between sites.*

**Lastly, the Discussion is largely devoted to the value of time-series, this could probably be condensed. As an edition, the results should be discussed in terms in the context of other studies of kelp forest or coastal variability in general (some were mentioned in the Introduction). Do these data fall within the range of biochemical observations made previously in other kelp forests?**
*We will look for ways to condense the Discussion and add a short paragraph comparing our dataset to the limited available data sets.*

**Other and Technical Comments:**

**Shorten the LPJPSMR acronym**
*We will remove this acronym*

**2.1 L23: of kelp of the kelp**
*We will correct this error*

**2.4 L6: provide reason or reference for phosphate assumption**
*We made phosphate measurements from February-August 2014. We did not discuss these in the manuscript, but they are included in the data set provided in the supplementary material. We will indicate that our phosphate concentrations used for carbonate system calculations come from our own measured, but not presented, data and that the reader can refer to the supplementary material to see the data.*

**2.4 L8: pHT is defined but not used in subsequent reporting of pH values in the Results**.
*We will remove the single reference to $pH_T$ as the Methods section already indicates that all pH data is on the total scale at in-situ temperature.*

**2.4 L11: double ))**
*The double )) was used because the reference was cited within a parenthetical statement. We will use brackets for the reference inside of the parenthetical statement.*

**For all time series figures: simplify x-axis date labels. Adding 01 as the day is not necessary and adds clutter. I recommend to simply label months as 1-12.**
*We will simplify the x-axis labels on the time series plots to only show the month*

**3.2 L23: "causing water column temperature differences of up to 4_C" add across what range of depths**
*We will include the depth of the Protected Offshore site in this sentence*

**3.3 L 16-17: "Surface DIC concentrations were generally much more spatially homogeneous than bottom water DIC concentrations." Could just be function of depth.**
*We agree that much (but not all) of the bottom DIC differences can be attributed to depth. We still think that much of the surface homogeneity is due to the strong biological influence in the surface waters, as we discuss in the Discussion section.*

**3.3 L25: has should be had**
*We will correct this error*

**3.3 L27: regarding pCO2 undersaturation, add "with respect to the atmosphere" if that is what you are measuring saturation against.**

*We will add this clause*

**Table 2: Since bottom depth differs across sites (7.5-16 m) and there are obvious depth effects, add the depth in m in () following the listed value in Table 2.**
*Depths for all study sites are already listed in Table 1*

**3.4 L21: "drove large variations in the ability to buffer against ongoing ocean acidification". Ocean acidification is not detectable across such a short time-series. Reword to simply say, "drove large variations in the Revelle Factor".**
*We will replace the sentence with the reviewer's suggested sentence*

**3.5: Why was aragonite used here (and not TA or DIC)?**
*We chose to use the aragonite saturation state because we thought this carbonate system parameter would be more familiar and more easily comprehendible to a broad suite of scientists working on ocean acidification and global change. We note that section 3.4 establishes that DIC variability is the dominant driver of observed variability in aragonite saturation state.*

**4.1 L28: Regarding this paragraph, it would be nice if you could find a reference showing the seasonality of phytoplankton blooms of this site. I imagine it is offset from the kelp forest growing season.**
*This would be a nice addition, but we do not have the data. As we identify in the Discussion section, this is a hypothesis for future work and points to the need to have an offshore control site against which to compare the nearshore biogeochemical variability.*

**4.2 L22 - State the actual findings. The largest source of variation seems to be the protected vs. unprotected sites, which is actually a function of the oceanographic features, not a function of the biology of the kelp forest. The biological control is in this study is the depth gradient (where primary production takes up DIC at the surface).**
*We hypothesize that oceanographic processes control the bottom waters, but that biological processes control the surface waters (at least during periods of high primary production). We discuss this hypothesis, along with the supporting observations, in Section 4.1. The paragraph that you reference serves as a starting point for discussing the implications of the work. Further discussion of the mechanisms behind the observed variability would be redundant from the previous section.*

**Pg. 11 L9: inconsistent use of OA vs. ocean acidification. I recommend to not use the acronym at all.**
*We will remove all uses of 'OA' and replace with 'ocean acidification'*

**Pg 11, L10: other studies have shown this previously also: pH sensor-based studies in coastal environments but also cruise data for offshore regions. Cite references in support of this conclusion.**
*We respectfully disagree with the comment to include offshore references. This study is explicitly about a nearshore ecosystem. We already openly acknowledge the limitations of not having offshore data in our study. Discussing other's offshore observations of primary production is outside the scope of this manuscript.*

**Pg. 11 L13: The ocean is not acidic, acidifying and acidic are different.**
*We will replace with 'acidifying'.*

**Pg. 11 L16: Same as previous comment. It makes more sense to use 'low pH' instead of 'high acidity'**
*We will replace 'high acidity' with 'low pH'*
**This manuscript nicely describes spatial and temporal variation in carbon system variables in a kelp forest in Central California. The data presented are the first to report high frequency measurements of carbon system variables made at small spatial scales across an entire annual cycle within a kelp forest. The data reveal substantial depth-dependent, spatial, and seasonal differences. The authors suggest mechanisms that could be responsible for creating the observed variation. Sampling design, sample collection, and analytical methods all appear appropriate to the research questions posed. The organization of the paper is logical, the writing is clear, and the graphics are appropriate and informative. Below I offer specific comments intended to strengthen the manuscript.**

*Thank you for your thorough and considerate review of this paper. We address your specific comments below.*

**Page 2, line 8: more clearly stated as "Calcification and dissolution of kelp-associated organisms, especially shelled invertebrates, can modify water chemistry. . ."**
*Our current sentence "Calcification and dissolution modify the water chemistry through the uptake and release of carbonate and bicarbonate ions, which modify the total alkalinity and dissolved inorganic carbon" provides a more complete mechanistic description of the water chemistry changes due to calcification and dissolution than does the suggested sentence. We prefer to keep the main components of our sentence, but will look into whether the sentence can be re-worded to be more concise.*

**Page 2, line 16: better stated as " Despite the recognized importance of the kelp forest. . ."**
*We will use this wording in the revised manuscript*

**Page 2, line 16: I searched but did not find reference to kelp as an "ecosystem architect". The earlier use of the term "foundation species" is more consistent with the ecological literature.**
*We will replace "ecosystem architect" with "foundation species" in the revised manuscript*

**Page 2, lines 25-27: Important points are made here. It would be helpful to clearly return to these in the discussion section.**
*We do return to these points in section 4.2. We will add additional text to section 4.2 to more clearly indicate our return to these discussion points.*

**Section 2.4 Satellite derived estimates: Estimating kelp canopy biomass is notoriously difficult. The authors have done a good job estimating relative changes in biomass over time but I found no indication in the text or figures to indicate error in this estimate. The inclusion of error estimates would be helpful.**
*We estimated giant kelp biomass using empirical relationship between diver estimated kelp canopy biomass and Landsat kelp fraction. We have generated 95% confidence intervals about this relationship at each time point, for each site, and will include these error estimates in the revised manuscript.*

**Section 3.4: Carbon systems variables differ between surface and bottom, consistent with the intrusion of CO2 enriched water at bottom and photosynthetic activity at the surface. Here or in the discussion it could be helpful to mention that the observed surface-to-bottom variation suggests that benthic calcifiers appear neither to be influencing TA nor do they appear to be benefitting from the effects of photosynthesis on water chemistry, which seem to be confined to surface waters. Moreover, understory seaweeds, which can achieve substantial biomass in kelp forests, don't appear to affect water chemistry appreciably (tho this was not tested). A fuller discussion of these considerations could be helpful.**
*Thank you for bringing these points up. We will expand section 4.1: 'Mechanisms of observed variability' to include the discussions of TA variability, or lack thereof, as well as understory production.*

**Section 4.2: The discussion of "space-for-time substitutions" is reasonable, but in my opinion is less compelling than other arguments that can be made concerning kelp forest ecosystems in an era of global change. I'd encourage the authors to open the discussion with the most compelling inferences that can be drawn from their data.**

*We respectfully disagree with the reviewer on this point. We believe that the most impactful results from our study point to the importance of considering small-scale spatial variability in kelp forests. Small-scale spatial variability potentially creates differential organismal responses to local conditions, which may then scale to create observed ecosystem-level responses to environmental conditions. Therefore, understanding the organismal response to small-scale variability is critical to predicting the ecosystem response. The variations in DIC create opportunities for looking at how organismal responses differ between high and low DIC regions of the kelp forest. This "space-for-time" approach encompasses observational and manipulative work that will help build our understanding from small-scale spatial variability to organismal response to ecosystem response.*

**Page 11, lines 10-18: The discussion of refugia could be refined. Assuming that photosynthesis within the canopy modulates stress due to high CO2/low pH, it's difficult to think of very many organisms (especially calcifying organisms) that can take advantage of this. These are likely to be limited to epibionts on kelp blades and perhaps a few canopy-associated fish species. A much larger number of calcifying taxa are associated with the benthos, where water conditions are likely to be less conducive to calcification and growth when omega is low. Consequently, the potential refugium created by the canopy is spatially unassociated with the bulk of benthic species. Moreover, the persistence of refugia in such a dynamic system is questionable.**

*We agree that benthic organisms are unlikely to benefit from any biogeochemical refugia created by the photosynthetic activity in the kelp canopy. However, our results point to consistent vertical biogeochemical gradients during times of high kelp biomass. Therefore, any mobile organisms sensitive to high $CO_2$ may increasingly find refuge in the surface waters. Testing this idea is beyond the scope of our paper (and we don't have any data with which to do so). However, we believe it is important to mention this hypothesis in the Discussion section so that others can use our data as motivation to more formally test for the presence and importance of biogeochemical refugia in such dynamic systems.*

**Page 11, lines 25-34: Comments about water quality criteria are reasonable: for instance, it's important to point out that the variability observed in this study exceeds that of water quality criteria now in existence. However, the paragraph doesn't seem particularly nuanced, beyond references to Boehm et al, Weisberg et al, and Chan et al. I encourage the authors to more fully consider the implications of their data for water quality criteria.**

*We use this paragraph to compare our observations to existing water quality regulations. However, we believe that an extensive review of water quality regulations along the west coast of North America, and how they interact with biogeochemical variability in a kelp forest, is beyond the scope of this paper. Extensive additions to this section of the paper would further extend the Discussion section of the paper. Reviewer #1 already suggests reducing the length of the Discussion and we believe further lengthening would take focus away from the main scientific points of the paper.*

**Page 11, line 28: replace "supervising agencies" with "regulatory agencies".**

*We will replace "supervising agencies" with "regulatory agencies" in the revised manuscript*

**Page 12, line 8: distinguish between "fully constrained" and "over-constrained" with respect to carbon system variables.**

*Thank you for identifying this subtle, but very important, difference. We will clarify our language here to emphasize the crucial importance of fully constraining the carbonate system (measuring 2 carbonate system parameters), with a goal of over-constraining the carbonate system (measuring 3 or more carbonate system parameters).*

**Page 12, line 18: replace "harvesting" with "kelp harvesting".**
*We will replace "harvesting" with "kelp harvesting" in the revised manuscript*

**Page 12, line 20: should read "effects of the kelp canopy".**
*We will use "effects of the kelp canopy" in the revised manuscript*

**Page 12, line 22: replace "chemically homogenize" with "reduce gradients in".**
*We will replace "chemically homogenize" with "reduce gradients in" in the revised manuscript*

---

## Author Response (AR2)

10 Received and published: 4 October 2016

General Comments:

This study presents 1+ year time-series data of weekly samples of carbonate chemistry across a small spatial scale of a kelp forest covering two summer seasons. The data include surface and bottom samples in exposed and protected sites and from inside to outside the kelp forest. The data are of extremely high quality. The

15 paper is well written, articulate, and has logical organization with nice transitions. While carbonate chemistry time series papers are increasing in number, this paper contributes novel and valuable data on small scale spatial variability (depth and spatial). In support of publishing this paper, I consider my comments as minor revisions which would improve the scientific quality from 'good' to 'excellent'.

20 Thank you for your thoughtful comments and careful review of this paper. We address your specific comments below.

**Specific Comments:**

I have three specific comments, two with regard to the spatial variability. First, bottom water sampled by site is confounded by depth, which is not explicitly addressed. The spatial variation of bottom water could just be an

- 25 artifact of the stratification of the water column within which the kelp forest sits (deeper waters have more DIC, so therefore bottom waters of deeper sites will have lower DIC values than bottom waters of shallower sites). The potential depth dependency of the observed dynamics (and conclusions) should be addressed and contextualized with the aims of the study (and the sampling design of surface and bottom waters, which was not explained). The data are valuable in terms of understanding the variation of what, for instance, a benthic
- 30 kelp forest inhabitant might experience, but then that perspective should be included (Introduction and Discussion).

We agree that bottom water sample variability is confounded by depth variability. Sections 3.4 and 3.5 already address the role of depth in establishing bottom water carbon system variability. Section 4.1 ("Mechanisms of observed variability") has been edited to include more text and explicit discussion of the role of depth variability in shaping the observed spatial

35 patterns in carbonate chemistry. However, as we also discuss in Section 4.1, differences in bottom water exchange between

the protected and exposed sides (as measured by ADCPs at similar depths), suggest additional localized factors that contribute to spatial variability. Finally, we have added a sentence to Section 2.2 to provide context for the sampling depths we chose in this study.

- 5 Second, the most valuable portions of this study are the depth gradient (well developed and presented) and the spatial gradient of the time-series (from inside to outside a kelp forest, exposed to protected). The presentation of the latter (Section 3.5) is extremely short and the figures comprise mostly of statistical numbers and not meaningful observations. The authors do themselves a disservice by not highlighting this aspect of the study more in depth. Figures 9 and 10 do not contribute anything that could not be shown in a
- 10 table (Fig. 9b, 8, and S8 display duplicate data in every plot). Fig. 9a could be interesting if shown as a line graph (bar graph is too cluttering) but I don't think it's necessary in the first place. Instead, I was expecting a figure showing the gradient in carbonate chemistry from inside to outside the kelp forest at the two contrasting sites (protected, exposed). How does these gradient change by season? It looks like the largest spatial differences occur between the exposed vs. protected site and not within the inside vs. outside (I
- 15 suspect that differences between inside and outside kelp forest will only be apparent with higher frequency sampling). The statistics show this, but the figure space would be better used by using the real data (e.g. select parts of the time-series, moving averages, etc.).

We reorganized Figure 9. Figure 9b in the Biogeosciences Discussion paper is now Fig. 9a, which helps highlight the clustering of the Pro Inn and Pro Mid sites relative to the other four sites. Figure 9a in the Biogeosciences Discussion paper

- 20 is now Fig. 9b and has been modified to show the histogram of aragonite saturation state observations for two groupings: Pro Inn + Pro Mid and the other four sites. Grouping the data into two sets helps better illustrate the spatial differences between Pro Inn and Pro Mid relative to the four deeper sites. We added additional text to Section 3.5 to better complement the new Figs. 9a and 9b, as well as to better clarify the unique and complementary contributions of Figs. 9 and 10. Specifically, Fig. 9 shows differences in average bottom conditions, whereas Fig. 10 shows differences in bottom co-
- 25 variation between sites.

Lastly, the Discussion is largely devoted to the value of time-series, this could probably be condensed. As an edition, the results should be discussed in terms in the context of other studies of kelp forest or coastal variability in general (some were mentioned in the Introduction). Do these data fall within the range of biochemical observations made previously in other kelp forests?

We have added several sentences to the beginning of the Discussion section to contextualize our results by comparing our pH range to that of the southern California studies listed in the Introduction.

2

Other and Technical Comments:

35

**Shorten the LPJPSMR acronym**

We have removed this acronym

**2.1 L23: of kelp of the kelp**

5 We corrected this error

**2.4 L6: provide reason or reference for phosphate assumption**

We made phosphate measurements from February-August 2014. We have edited the sentence in question to let readers know that these measurements were made during the study and we refer them to the attached data file for the original measurements.

**2.4 L8: pHT is defined but not used in subsequent reporting of pH values in the Results.**

We removed references to  $pH_T$

**15 2.4 L11: double ))**

10

The double )) was used because the reference was cited within a parenthetical statement. We replaced the inside set of parenthesis with brackets for the reference inside of the parenthetical statement.

**For all time series figures: simplify x-axis date labels. Adding 01 as the day is not**

**20 necessary and adds clutter. I recommend to simply label months as 1-12.**

We have simplified the x-axis labels on all time series plots to show only the month number.

**3.2 L23: "causing water column temperature differences of up to 4\_C" add across what range of depths**

25 We added the depth of the Protected Offshore site in this sentence

**3.3 L 16-17: "Surface DIC concentrations were generally much more spatially homogeneous than bottom water DIC concentrations." Could just be function of depth.**

We agree that much (but not all) of the bottom DIC differences can be attributed to depth. We still think that much of the

30 surface homogeneity is due to the strong biological influence in the surface waters, as we discuss in the Discussion section.

**3.3 L25: has should be had**

We corrected this error

**35 3.3 L27: regarding pCO2 undersaturation, add "with respect to the atmosphere" if that**

**is what you are measuring saturation against.**

We added this clause

5

**Table 2: Since bottom depth differs across sites (7.5-16 m) and there are obvious depth**

effects, add the depth in m in () following the listed value in Table 2.

Depths for all study sites are already listed in Table 1

**3.4 L21: "drove large variations in the ability to buffer against ongoing ocean acidification".**

**Ocean acidification is not detectable across such a short time-series. Reword to simply say, "drove large 10 variations in the Revelle Factor".**

We replaced the sentence with the reviewer's suggested sentence

**3.5: Why was aragonite used here (and not TA or DIC)?**

We chose to use the aragonite saturation state because we thought this carbonate system parameter would be more familiar and more easily comprehendible to a broad suite of scientists working on ocean acidification and global change. We note that section 3.4 establishes that DIC variability is the dominant driver of observed variability in aragonite saturation state.

**4.1 L28: Regarding this paragraph, it would be nice if you could find a reference showing the seasonality of phytoplankton blooms of this site. I imagine it is offset from the**

20 kelp forest growing season.

This would be a nice addition, but we do not have the data. As we identify in the Discussion section, this is a hypothesis for future work and points to the need to have an offshore control site against which to compare the nearshore biogeochemical variability.

- 25 4.2 L22 State the actual findings. The largest source of variation seems to be the protected vs. unprotected sites, which is actually a function of the oceanographic features, not a function of the biology of the kelp forest. The biological control is in this study is the depth gradient (where primary production takes up DIC at the surface).
- We hypothesize that oceanographic processes control the bottom waters, but that biological processes control the surface waters (at least during periods of high primary production). We discuss this hypothesis, along with the supporting observations, in Section 4.1. The paragraph that you reference serves as a starting point for discussing the implications of the work. Further discussion of the mechanisms behind the observed variability would be redundant from the previous section.
- 35 Pg. 11 L9: inconsistent use of OA vs. ocean acidification. I recommend to not use the

**acronym at all.**

We have removed all uses of 'OA' and replaced with 'ocean acidification'

**Pg 11, L10: other studies have shown this previously also: pH sensor-based studies**

**5 $\,$ in coastal environments but also cruise data for offshore regions. Cite references in**

**support of this conclusion.**

We respectfully disagree with the comment to include offshore references. This study is explicitly about a nearshore ecosystem. We already openly acknowledge the limitations of not having offshore data in our study. Discussing other's offshore observations of primary production is outside the scope of this manuscript.

10

**Pg. 11 L13: The ocean is not acidic, acidifying and acidic are different.**

We replaced with 'acidifying'.

**Pg. 11 L16: Same as previous comment. It makes more sense to use 'low pH' instead**

15 of 'high acidity'

We replaced 'high acidity' with 'low pH'

**Interactive comment on "A year in the life of a central California kelp forest: physical and**

20 biological insights into biogeochemical variability" by David A. Koweek et al.

Anonymous Referee #2

Received and published: 10 October 2016

This manuscript nicely describes spatial and temporal variation in carbon system variables in a kelp forest in Central California. The data presented are the first to report high frequency measurements of carbon system

25 variables made at small spatial scales across an entire annual cycle within a kelp forest. The data reveal substantial depth-dependent, spatial, and seasonal differences. The authors suggest mechanisms that could be responsible for creating the observed variation. Sampling design, sample collection, and analytical methods all appear appropriate to the research questions posed. The organization of the paper is logical, the writing is clear, and the graphics are appropriate and informative. Below I offer specific comments intended to

30 strengthen the manuscript.

Thank you for your thorough and considerate review of this paper. We address your specific comments below.

Page 2, line 8: more clearly stated as "Calcification and dissolution of kelp-associated organisms, especially shelled invertebrates, can modify water chemistry..."

The sentence now reads: "Calcification and dissolution of kelp-associated organisms, especially shelled invertebrates, modify the water chemistry through the uptake and release of carbonate and bicarbonate ions, which modify the total alkalinity (TA) and DIC."

5 Page 2, line 16: better stated as " Despite the recognized importance of the kelp forest. . ."

Opening clause in the paragraph now reads as "Despite the recognized importance of kelp as a foundation species and biogeochemical agent,..."

Page 2, line 16: I searched but did not find reference to kelp as an "ecosystem architect". The earlier use of the 10 term "foundation species" is more consistent with the ecological literature.

We replaced "ecosystem architect" with "foundation species"

Page 2, lines 25-27: Important points are made here. It would be helpful to clearly return to these in the discussion section.

15 We have added text to the opening sentence of section 4.2 to more clearly indicate our return to these discussion points.

Section 2.4 Satellite derived estimates: Estimating kelp canopy biomass is notoriously difficult. The authors have done a good job estimating relative changes in biomass over time but I found no indication in the text or figures to indicate error in this estimate. The inclusion of error estimates would be helpful.

- 20 We added text to Section 2.5 (was incorrectly labelled as Section 2.4 earlier) to describe the error estimation procedure for the kelp canopy biomass estimates. Figure 2a now shows the 95% confidence interval around the kelp canopy biomass estimates.
- Section 3.4: Carbon systems variables differ between surface and bottom, consistent with the intrusion of CO2 enriched water at bottom and photosynthetic activity at the surface. Here or in the discussion it could be helpful to mention that the observed surface-to-bottom variation suggests that benthic calcifiers appear neither to be influencing TA nor do they appear to be benefitting from the effects of photosynthesis on water chemistry, which seem to be confined to surface waters. Moreover, understory seaweeds, which can achieve substantial biomass in kelp forests, don't appear to affect water chemistry appreciably (tho this was not tested). A fuller discussion of these considerations could be helpful.

We expanded section 4.1: 'Mechanisms of observed variability' to include a brief discussion of how we do not see strong evidence for calcification, as inferred from the TA data, and that understory production may be an important process, but we do not have the vertical sampling resolution to quantify the role of understory production in the observed biogeochemical gradients.

Section 4.2: The discussion of "space-for-time substitutions" is reasonable, but in my opinion is less compelling than other arguments that can be made concerning kelp forest ecosystems in an era of global change. I'd encourage the authors to open the discussion with the most compelling inferences that can be drawn from their data.

- 5 We respectfully disagree with the reviewer on this point. We believe that the most impactful results from our study point to the importance of considering small-scale spatial variability in kelp forests. Small-scale spatial variability potentially creates differential organismal responses to local conditions, which may then scale to create observed ecosystem-level responses to environmental conditions. Therefore, understanding the organismal response to small-scale variability is critical to predicting the ecosystem response. The variations in DIC create opportunities for looking at how organismal
- 10 responses differ between high and low DIC regions of the kelp forest. This "space-for-time" approach encompasses observational and manipulative work that will help build our understanding from small-scale spatial variability to organismal response to ecosystem response.
- Page 11, lines 10-18: The discussion of refugia could be refined. Assuming that photosynthesis within the canopy modulates stress due to high CO2/low pH, it's difficult to think of very many organisms (especially calcifying organisms) that can take advantage of this. These are likely to be limited to epibionts on kelp blades and perhaps a few canopy-associated fish species. A much larger number of calcifying taxa are associated with the benthos, where water conditions are likely to be less conducive to calcification and growth when omega is low. Consequently, the potential refugium created by the canopy is spatially unassociated with the 20 bulk of benthic species. Moreover, the persistence of refugia in such a dynamic system is questionable.
- We agree that benthic organisms are unlikely to benefit from any biogeochemical refugia created by the photosynthetic activity in the kelp canopy. However, our results point to consistent vertical biogeochemical gradients during times of high kelp biomass. Therefore, any mobile organisms sensitive to high  $CO_2$  may increasingly find refuge in the surface waters. Testing this idea is beyond the scope of our paper (and we don't have any data with which to do so). However, we believe it
- 25 is important to mention this hypothesis in the Discussion section so that others can use our data as motivation to more formally test for the presence and importance of biogeochemical refugia in such dynamic systems.

Page 11, lines 25-34: Comments about water quality criteria are reasonable: for instance, it's important to point out that the variability observed in this study exceeds that of water quality criteria now in existence. However, the paragraph doesn't seem particularly nuanced, beyond references to Boehm et al, Weisberg et al, and Chan

30

et al. I encourage the authors to more fully consider the implications of their data for water quality criteria. We use this paragraph to compare our observations to existing water quality regulations. However, we believe that an extensive review of water quality regulations along the west coast of North America, and how they interact with biogeochemical variability in a kelp forest, is beyond the scope of this paper. Extensive additions to this section of the paper

would further extend the Discussion section of the paper. Reviewer #1 already suggests reducing the length of the Discussion and we believe further lengthening would take focus away from the main scientific points of the paper.

**Page 11, line 28: replace "supervising agencies" with "regulatory agencies".**

5 We replaced "supervising agencies" with "regulatory agencies"

**Page 12, line 8: distinguish between "fully constrained" and "over-constrained" with respect to carbon system variables.**

Thank you for identifying this subtle, but very important, difference. We revised this sentence to emphasize the crucial

10 importance of fully constraining the carbonate system (measuring 2 carbonate system parameters), with a goal of overconstraining the carbonate system (measuring 3 or more carbonate system parameters).

8

**Page 12, line 18: replace "harvesting" with "kelp harvesting".**

We replaced "harvesting" with "kelp harvesting"

15

**Page 12, line 20: should read "effects of the kelp canopy".**

We now use "effects of the kelp canopy"

**Page 12, line 22: replace "chemically homogenize" with "reduce gradients in".**

20 We replaced "chemically homogenize" with "reduce gradients in"

**A year in the life of a central California kelp forest: physical and biological insights into biogeochemical variability**

David A. Koweek1,\*, Kerry J. Nickols2,3, Paul R. Leary2, Steve Y. Litvin2, Tom W. Bell4, Timothy Luthin1, Sarah Lummis2,#, David A. Mucciarone1, and Robert B. Dunbar1

Correspondence to: David A. Koweek (dkoweek@carnegiescience.edu)

**15 Abstract.**

95060, USA

[revised manuscript text omitted]

David Koweek 11/28/2016 12:29 PM Formatted: Font:(Default) Times New Roman, 10 pt, Not Bold, Not Highlight

David Koweek 11/28/2016 12:02 PM Formatted: Not Superscript/ Subscript David Koweek 11/28/2016 12:03 PM Deleted: NBS David Koweek 11/28/2016 12:03 PM Formatted: Not Superscript/ Subscript David Koweek 11/28/2016 12:02 PM Deleted: T David Koweek 11/28/2016 11:47 AM Deleted: Despite kelp forest's recognized importance David Koweek 11/28/2016 11:48 AM Deleted: n ecosystem architect

**2. Methods**

**2.1 Study Site**

Our study site is located along the eastern side of the Monterey Peninsula in the central portion of the California Current Large Marine Ecosystem (Fig. 1) (Checkley and Barth, 2009). This region is characterized by seasonal upwelling

- conditions driven by increased northwesterly winds from ~March-September (Checkley and Barth, 2009) (Fig. S1). 5 Although much of the Monterey Peninsula is protected from the stronger along shore winds and associated surface conditions experienced by the nearby exposed coastline and northern Monterey Bay, it still experiences cross-shore transport associated with upwelling (Woodson, 2013). Advection of upwelled waters and the local topography also facilitate the propagation of internal bores into southern Monterey Bay, which act as an additional mechanism for introducing dense, high
- CO2 deep ocean water into nearshore habitats and driving variability in CO2 chemistry at short temporal scales (Booth et al., 10 2012; Walter et al., 2014).

The kelp forest is located in the Lovers Point-Julia Platt State Marine Reserve, directly offshore of Hopkins Marine Station along the wave-protected side of the Monterey Peninsula (Fig. 1). The reserve was created in 2007 as part of the network of central California marine protected areas designated under the California Marine Life Protection Act (California

15 Department of Fish and Wildlife 2016). The kelp forest in the Lovers Point-Julia Platt State Marine Reserve, has been protected since 1931, originally as the Hopkins Marine Life Refuge. Due to its protected status, the reserve serves as a natural laboratory to investigate the biogeochemistry of a central California kelp forest featuring low levels of human disturbance.

[revised manuscript text omitted]

David Koweek 11/28/2016 11:43 AM **Deleted:** ability to buffer against ongoing ocean acidification

David Koweek 11/28/2016 11:41 AM Deleted: s

over the annual cycle (Fig. S7), which nearly spans the range of global surface ocean values from offshore waters (Sabine et al., 2004). Following other carbon system properties, the highest and most variable RFs were observed in the bottom waters during spring and summer. Photosynthetic uptake in the surface during upwelling months lowered RFs to between 10-12, with occasional values reaching below 10. Winter RFs for surface and bottom water samples converged around 12-14.

**5 3.5 Spatial Variability**

Full-year histograms of  $\Omega_{Ar}$  in kelp forest bottom waters revealed significant differences (Wilcoxon rank sum test,  $\alpha$ =0.05) between the sites generally according to site depth and orientation (Fig. 9). The Protected Middle and Protected Inside sites had significantly higher  $\Omega_{Ar}$  than the other four sites (except in the pairwise test between Protected Middle and Exposed Middle), which further highlights the clustering of the Protected Middle and Protected Inside sites relative to the

other sites (Fig. 9). There were no site-to-site significant differences in surface  $\Omega_{Ar}$  values, indicating greater homogeneity at 10 the surface. Not only were there significant differences in bottom water  $\Omega_{\Lambda_{\Psi}}$  but the bottom waters were spatially decoupled as well. Throughout the year, bottom,  $\Omega_{Ar}$  measurements from Protected Inside and Protected Middle were far less correlated with measurements from the deeper and exposed sites, as compared to the correlations among the deeper, more exposed sites (Fig. 10). In contrast, the surface  $\Omega_{Ar}$  was highly correlated between all six sites (Fig. 10). The bottom water decoupling and surface water coupling were largely maintained during periods of both strong and weak upwelling (Fig. S8).

15

**4. Discussion**

This study demonstrates strong spatial and temporal variability in CO2 system chemistry within a central California kelp forest using nearly 800 DIC and TA observations that ranged across six sites, multiple depths, and spanning a period of 14 months. This data set represents one of the largest multi-parameter CO2 system data sets completed in a kelp forestyOur 20 spatially expansive sampling captured larger pH ranges than in many of the southern California pH studies. Our calculated range in pH of 7.7-8.33 was larger than the range observed in kelp forests from single pH sensors in the Channel Islands (Kapsenberg and Hofmann, 2016; 7.88-8.12), the La Jolla kelp forest (Takeshita et al., 2015; ~7.78-8.12), and the Santa Barbara Channel (Hofmann et al., 2011; ~7.7-8.25). Our pH range was slightly smaller than the range of ~7.65-8.39 observed by Frieder et al (2012) in the La Jolla kelp forest using a network of pH sensors, although the high temporal 25 resolution of the sensors captured event-scale variability that would likely be missed by the weekly sampling alone in our

study. We now consider the physical and biological drivers of our observed variability and the implications of our work for understanding kelp forests in an era of global change.

**4.1 Mechanisms of observed variability**

Co-variation of water column structure and kelp canopy biomass provided evidence for the strong influence of regional scale upwelling processes on the seasonal cycle of the kelp canopy (Fig. 2a). Bottom water advection during 30

17

**David Koweek 12/1/2016 5:12 PM Deleted: carbonate chemistry David Koweek 12/1/2016 5:23 PM Deleted: water David Koweek 12/1/2016 5:19 PM Deleted: the protected shallower sites David Koweek 12/1/2016 5:24 PM Deleted: was **Deleted:** relative to $\Omega_{Ar}$ of surface waters (Fig. 10). This decoupling 12/12/2016 12:15 PM David Kow Deleted: as**

David Koweek 11/30/2016 3:46 PM Deleted: and provides critical baseline data for

detecting biogeochemical change in this system

David Koweek 11/30/2016 5:16 PM Deleted: ( ek 11/30/2016 5:16 PM David Kowe Deleted: , David Koweek 11/30/2016 5:33 PM Deleted: the David Koweek 11/30/2016 5:34 PM Deleted: this

[revised manuscript text omitted]

In contrast to the evidence for surface photosynthesis, we do not see biogeochemical evidence for significant calcification within the kelp forest. Benthic calcification would have depleted TA in the bottom waters relative to the surface, yet we see no consistent pattern in the vertical TA gradients (Fig. 6b). The lack of TA gradients suggests that the biogeochemical signature of calcification in the bottom waters of the kelp forest was small relative to bottom water exchange in the kelp forest.

**4.2 Implications for understanding kelp forest ecosystems in an era of global change**

- 10 Our data provide critical biogeochemical baselines to assess kelp forest responses to local and global stressors, and further highlight the necessity of long-term, spatially expansive sampling for gaining insights into the patterns and controls on kelp forest biogeochemistry. Kelp forests are spatially and temporally dynamic environments generally found along upwelling margins, so any assessment of their biogeochemical variability must account for the variability in their seasonalto-interannual physical (Bograd et al., 2009; Checkley and Barth, 2009) and biological controls (Bell et al., 2015b). We have
- 15 demonstrated how variations in these physical and biological controls in a central California kelp forest create biogeochemically heterogeneous environments. This creates the opportunity to do localized "space-for-time" experiments to understand benthic organismal and community response to high CO2 seawater. In contrast to larger space-for-time experiments (Hofmann et al., 2014), a single kelp forest may be sufficient for replicating the geochemical gradients expected between current conditions and future conditions under ocean acidification. These localized space-for-time experiments
- 20 could be conducted in small environments (hundreds of meters, not hundreds of kilometers) with near identical biological assemblages and for a fraction of the cost and effort. Combining transplant experiments with observational work could leverage the biogeochemical differences between areas of a kelp forest to yield important insights into the adaptive potential of benthic community members under the combined stresses of global change.
- David Koweek 11/30/2016 1:41 PM Deleted: with David Koweek 11/28/2016 11:36 AM Deleted: OA David Koweek 11/30/2016 1:42 PM Deleted: conditions

Aqueous CO2 variability typically does not occur in isolation of other biogeochemical and environmental changes. 25 Although we do not have the O2 measurements to accompany our carbon system time series, co-variation of CO2 and O2 has been well documented in California's coastal ecosystems (Booth et al., 2012; Frieder et al., 2012; Takeshita et al., 2015) and frequent episodic hypoxia has been documented near our study site (Booth et al., 2012). While a large component of this variation is naturally occurring along upwelling margins, low O2/high CO2 events are predicted to increase in frequency in the future due to oxygen minimum zone expansion (Bograd et al., 2008; Booth et al., 2014; Stramma et al., 2010) and

30 increases in upwelling favorable winds (Bakun, 1990; Sydeman et al., 2014). Understanding the co-variation, or lack thereof, between critical environmental parameters (e.g., temperature, CO2 system variables, and O2) through long-term spatially expansive measurements is necessary to define the range and timescales of environmental conditions experienced by kelp forest inhabitants. Quantifying these environmental conditions can inform the design of more realistic laboratory

experiments, featuring proper ranges and timescales of biogeochemical and thermal variability as opposed to chemostatic conditions (Reum et al., 2015). More realistic experimental conditions will yield additional insights into organismal and community response to climate change and OA.

Our data demonstrate that, despite the strong influence of physical processes, primary production can alter local

- 5 biogeochemistry. Understanding the role of foundation species, such as giant kelp, in creating biogeochemical refugia both through metabolic activity and alteration of the hydrodynamic regime within the kelp forest via increased residence time (Rosman et al., 2007) warrants further attention in an acidifying ocean, as they present potential to mitigate some stress effects of low O2, high CO2 water. The combination of physical and biological processes may create natural refugia for certain sensitive organisms or, conversely, create particularly stressful conditions for others. Organisms that can confine
- 10 themselves to the upper water column may be able to use the locally created biogeochemical refuge to avoid low pH, whereas organisms that must use the entire water column will experience large ranges in carbonate chemistry (pH, pCO2, and  $\Omega_{Ar}$ ) that will only grow larger with continued acidification.

In order to effectively manage critical coastal ecosystems in the face of climate change and OA, resource managers require both monitoring data and a process-based understanding of biogeochemical variability in order to identify changing

- 15 environmental conditions and forecast ecosystem responses in kelp forests (Boehm et al., 2015). Such understanding is currently lacking. Along the US west coast, states currently use decades-old water quality criteria for assessing pH, including that the pH should not drop below 6.5 and/or that it should not deviate more than 0.2 units from natural conditions (Weisberg et al., 2016). Yet we have demonstrated changes up to 0.5 pH units within a single kelp forest in just 15 meters depth. Additional observational studies may prove useful in helping us refine our understanding of coastal water quality away from
- static indicators and towards a more dynamic understanding of the ranges and controls on coastal water quality in order to better differentiate natural and anthropogenic effects, as has been demonstrated for dissolved oxygen (Booth et al., 2014).
   We recommend that regulatory agencies revise existing regulations to incorporate this understanding of natural variability in carbonate chemistry into their regulatory frameworks, as well as to expand the suite of carbonate chemistry water quality variables beyond pH alone (Weisberg et al., 2016). Developing stronger biological criteria for carbonate chemistry ranges
- 25 and variability deemed necessary to preserve critical marine living resources would help align scientific and management priorities by focusing research efforts on quantifying these acceptable ranges of conditions (Chan et al., 2016; Weisberg et al., 2016).

Comprehensive, spatially expansive monitoring is fundamental to characterizing the ranges and timescales of carbonate system variability in highly productive coastal ecosystems, such as kelp forests. Maintaining monitoring networks along coastal ecosystems is a crucial step to developing an understanding of natural vs. anthropogenic influences in the coastal zone in support management and regulatory efforts (Strong et al., 2014). Recent advances in sensor technology, such as the SeapHOx (Bresnahan et al., 2014), may make monitoring far less laborious than the data collection efforts in this study. However, we strongly recommend that monitoring programs measure at least two carbonate system parameters (TA, DIC, pH, pCO2) so as to fully constrain the carbonate system (McLaughlin et al., 2015), with a goal of measuring three or

20

David Koweek 11/28/2016 11:46 AN Deleted: increasingly acidic

David Koweek 11/28/2016 11:45 AM Deleted: high acidity

David Koweek 11/28/2016 11:49 AM Deleted: supervising

David Koweek 11/28/2016 12:09 PM Deleted: incorporate David Koweek 11/28/2016 12:06 PM Deleted: multiple more parameters to over-constrain the system, Coupling SeapHOx sensors to autonomous pCO2 (as discussed in Martz et al.

2015) or newly developed autonomous TA sensors (Spaulding et al., 2014) may accomplish this task. Integrating observations from individual sites along the California coast into larger coordinated networks, such as the California-Current Acidification Network, will provide a more complete perspective on the spatial and temporal variability of the aqueous  $CO_2$

system and its controls (Chan et al., 2016; McLaughlin et al., 2015).

Long-term observational data sets are also crucial for improving our process-based understanding of coastal ocean biogeochemistry. Data from studies such as these can be used to parameterize the process-based hydrodynamic-biogeochemical models that are needed to predict coastal ecosystem responses to climate change and ocean acidification (Boehm et al., 2015; McLaughlin et al., 2015). These models can help identify climate change and/or ocean acidification

- 10 hotspots along coastal ecosystems in the future and help marine living resource managers with scenario planning so that they can direct resources accordingly (Strong et al., 2014). Similarly, marine resource managers must consider the biogeochemical implications of any management activities, such as kelp harvesting, which alter kelp canopy density. We have shown considerable CO2 uptake at our study site during periods of high kelp canopy cover. While we were not able to directly attribute the CO2 uptake to *Macrocystis pyrifera*, the direct and indirect effects of the kelp canopy are likely to increase vertical gradients in water chemistry and therefore activities that decrease canopy density are likely to reduce
  - gradients in the water column.

5

Kelp forest ecology has long been a focal point for marine ecologists. This study complements the long history of kelp forest community (Dayton, 1985; Graham et al., 2007) and disturbance ecology (Edwards, 2004; Edwards and Estes, 2006) by highlighting the dynamic biogeochemistry of kelp forests over an annual cycle. The few existing biogeochemical

- 20 studies of *Macrocystis pyrifera*-dominated kelp forest along the California coast have thus far occurred in southern California (Frieder et al., 2012; Kapsenberg and Hofmann, 2016; Takeshita et al., 2015). This study expands on the southern California studies by adding the first fully resolving CO2 system chemistry study in a central California kelp forest where seasonality is relatively stronger and kelp cover is more variable (Bell et al., 2015b; Checkley and Barth, 2009; Reed et al., 2011).
- 25 We hope that the large data set presented here motivates the inclusion of biogeochemistry as a new approach to understand kelp forest ecosystem function and the feedbacks between biology, chemistry, and physics in these dynamic systems. We note that this study would not have been possible without a large and concerted field effort to establish a spatially expansive sampling program that ranged over gradients in kelp density and wave exposure. Yet this sampling program was necessary to generate insights into the relevant scales of kelp forest biogeochemical variability presented in this
- 30 study, variability that could not be discerned from a single sampling point or sensor alone. We advocate for monitoring efforts to extend beyond single site measurements within a kelp forest. Doing so will enhance our understanding of the biological and physical mechanisms giving rise to the observed biogeochemical variability; information crucial to accurately predicting the response of kelp forest ecosystems to global change.

21

David Koweek 11/28/2016 12:08 PM Deleted: (McLaughlin et al., 2015)

David Koweek 11/28/2016 11:36 AM Deleted: OA David Koweek 11/28/2016 11:36 AM Deleted: OA

[revised manuscript text omitted]